# Influence of open-source virtual-reality based gaze training on navigation performance in Retinitis pigmentosa patients in a crossover randomized controlled trial

Alexander Neugebauer [1] *, Alexandra Sipatchin[1], Katarina Stingl[2], Iliya Ivanov[3], Siegfried Wahl[1,3]

**1** Institute for Ophthalmic Research, ZEISS Vision Science Lab, University of Tübingen, Tübingen, Germany, **2** Center for Ophthalmology, University Eye Hospital, University of Tübingen, Tübingen, Germany, **3** Carl Zeiss Vision International GmbH, Aalen, Germany

* alex.ngbr@uni-tuebingen.de

## Abstract

### Methods

A group of RP patients (n = 8, aged 20-60) participated in a study consisting of two 4-week-phases, both carried out by the same patient group in randomized order: In the 'training phase', participants carried out a Virtual-Reality gaze training for 30 minutes per day; In the 'control phase', no training occurred. Before and after each phase, participants were tasked to move through a randomized real-world obstacle course. Navigation performance in the obstacle course as well as eye-tracking data during the trials were evaluated. The study is registered at the German Clinical Trials Register (DRKS) with the ID DRKS00032628.

### Results

On average, the time required to move through the obstacle course decreased by 17.0% after the training phase, the number of collisions decreased by 50.0%. Both effects are significantly higher than those found in the control phase ($p < 0.001$ for required time, $p = 0.0165$ for number of collisions), with the required time decreasing by 5.9% and number of collisions decreasing by 10.4% after the control phase. The average visual area observed by participants increases by 4.41% after training, however the effect is not found to be significantly higher than in the control phase ($p = 0.394$).

### Conclusion

The performance increase over the training phase significantly surpasses the natural learning effect found in the control phase, suggesting that Virtual-Reality based gaze training can have a positive effect on real-world navigation tasks for patients with RP. The training is available as work-in-progress open-source software.

**Data Availability Statement:** All relevant data are within the manuscript and its Supporting information files.

**Funding:** This work was supported by the Deutsche Forschungsgemeinschaft (DFG) (URL: https://www.dfg.de/en/, Grant: DFG IV 167/5-1 to II), including support in the form of salary for the author A.N. In addition, the Carl Zeiss Vision International GmbH provided support in the form of salaries for I.I. and S.W. Both funders did not have any additional role in the study design, data collection and analysis, decision to publish, or preparation of the manuscript. The specific roles of these authors are articulated in the 'author contributions' section.

**Competing interests:** This work was done in an Industry-on-Campus-cooperation between the University of Tübingen and Carl Zeiss Vision International GmbH. Two of the authors, I.I. and S. W., are employees of Carl Zeiss Vision International GmbH. This does not alter our adherence to PLOS ONE policies on sharing data and materials. There are no competing interests related to employment, consultancy, patents, products in development, or marketed products.

## Introduction

Retinitis pigmentosa (RP) is a subset of inherited retinal diseases characterized by progressive loss of the Visual Field (VF) due to the degeneration of the retina [1–3]. This loss of the VF starts at the periphery or middle periphery and leads to blindness in the long-term progression of the degeneration [4, 5]. Other symptoms of RP include blurriness of sight, glare sensitivity, as well as night blindness [1, 6]. RP is estimated to occur in about 1 in 4000 people [5, 7–10].

The condition in which visual information can only be perceived in the center of the VF is known as "tunnel vision". It can have severe impact on the daily lives of those affected by RP [6], especially in visual tasks such as navigation and visual search. At the time of writing there is only one approved gene therapy for retinitis pigmentosa [11]. Despite good results in efficacy of this therapy also on the visual field, in the majority of the patients halting the progression is not consistently possible [6, 8]. It is therefore essential to explore other methods that can improve the visual capabilities of RP patients—and thus improve their quality of life.

One of these approaches is gaze training, which involves teaching patients how to adjust their gaze movements to compensate for their missing visual areas [12, 13]. For patients with limited VF, an important technique for this approach is the use of exploratory saccades. Exploratory saccades are rapid eye movements that help explore the visual environment by quickly shifting the point of fixation to new locations [14]. While these eye movements can not directly improve the biological health of the retina or increase the size of the "static" VF, i.e. the visual area that can be perceived at any one time, they can increase the visual area that is observed over time, facilitating the detection of new visual information. By incorporating exploratory saccades into gaze training, patients can learn to adapt their gaze movements to partly accommodate for their limited VF and observe larger areas around them. This can lead to better obstacle detection, safer navigation, and an overall higher level of independence in everyday visual tasks.

The concept of gaze training for low-vision compensation has been investigated and applied before. Nelles et al. [12] and Pambakian et al. [13] both evaluated the effects of a four-week supervised gaze training in patients with hemianopia, a condition of half-sided visual field loss. In the study of Nelles et al., training included specific instructions for adaptive gaze strategies, whereas patients in the study by Pambakian et al. were free to develop their own gaze strategies. In both studies, it could be shown that after gaze training, patients had a significantly shorter reaction time for visual stimuli in the non-seeing side of the visual field. Additionally, patients reported improvements in several vision-related quality of life aspects after training. Nguyen et al. [15], Roth et al. [16], and Ivanov et al. [14] conducted studies comprised of six weeks of unsupervised at-home training with a screen-based exploratory saccade training in patients with hemianopia (Roth et al.) and RP (Nguyen et al., Ivanov et al.), respectively. They were assessing the training effect on visual search (Roth et al., Ivanov et al.), scene exploration (Roth et al.), and the effect on real-world mobility (Nguyen et al., Ivanov et al.). Similarly, Kuyk et al. [17] investigated the effects of five days of visual search training on both search and real-world mobility tasks in people with different visual field impairments. All three studies with visual search testing paradigm found improvements in reaction time after training, both for digital feature search and for real-world object selection. For the real-world mobility tests, limited effects were reported: Nguyen et al. found significant training effects for real-world navigation in patients with visual field size <10°. In the study by Ivanov et al., RP patients displayed a significant improvement in walking speed, but no improvements in collision avoidance. In the study by Kuyk et al., no significant effects in walking speed were found, but collision avoidance improved in one of the two tested lighting conditions. A different

study by Hazelton et al. [18] compared the effectiveness of four different eye movement training tools on patients with stroke-induced visual field loss. Quantitatively, no significant improvements were found for any of the four tools, with only individual patients displaying improvements in certain testing paradigms such as visual search or reading speed. Qualitative assessment suggested, however, that patients perceived a positive influence of the training tools on everyday visual tasks. Gunn et al. [19] conducted a study in which patients with visual impairments caused by glaucoma underwent two supervised one-hour training sessions comprised of both general and task-specific gaze strategy training and instructions, including video showcases of 'expert' performers. Effects of the training were evaluated in a foot-placement task and a short obstacle avoidance task, with significant performance improvements found in foot placement accuracy and obstacle avoidance, though at a reduction in movement speed in the obstacle avoidance task. Additionally, changes in the patients' gaze behavior were registered after training. Lastly, Young and Holland [20] tested whether gaze training could improve mobility and reduce risk of falling even in elderly persons with no visual field impairment. After a supervised training in which participants received instructions on gaze behavior, participants were found to show increased foot placement accuracy, with no significant changes on movement speed. It can be noted that all of these training paradigms rely on either personal supervision and instructions (Nelles et al., Pambakian et al., Gunn et al., Young and Holland) or use a screen-based setup for at-home training (Nguyen et al., Roth et al., Ivanov et al., Kuyk et al., Hazelton et al.). With the constant advancements in technology and accessibility of Virtual Reality (VR) headsets, a question is raised about the potential of VR to be applied for gaze training purposes. Compared to conventional, computer display based setups, VR devices offer a number of possible advantages.

- The displays of a VR headset cover larger visual angles than a traditional computer screen, with most commercially available VR headsets featuring visual angles of 90˚ per eye or higher [21]. Assuming the recommended minimal distance from a working screen of 50cm [22], a 45" screen (99.7cm×56.0cm) is required to match the visual angle of a VR headset at least in horizontal dimension, and an 80" computer screen (177cm×99.6cm) would be required to also match the vertical visual angle.

- In addition, VR headsets can measure head rotations and adjust the displayed image in real-time to mimic the effect of "looking around". This further increases the visual angles at which VR headsets can display visual content, allowing for a full 360˚ view.

- Lastly, the use of VR allows the risk-free simulation of immersive, interactive 3D environments that provide a perspective and visual experience much closer to that of the real world.

To the best of our knowledge, at the point of writing there is no Virtual Reality based gaze training for people with visual field deficit apart from the one presented in this work. However, research on the use of Virtual Reality for adjusting gaze behavior in other fields, such as for industry task training [23], medical procedures [24], or as therapeutic intervention for patients with mental health disorders [25], suggests that the use of VR applications is feasible to influence gaze behavior. In addition, it has been shown that skills trained in VR can have sustained transfer effect to real-world performances in tasks such as tire changing [26], golfing [27], simulated electronic assembly [28], or walking with minimal foot clearance [29]. In this work, we are investigating the potential of Virtual Reality to be applied for unsupervised, at-home gaze training, as well as the influence of gaze training in a virtual environment on the navigation performance in real-world tasks.

## Materials and methods

The first part of this section will focus on the developed gaze training, its implementation, and the intentions behind its design. Subsequently, an experimental study will be presented to show how the gaze training impacted real-world navigation. A CONSORT flowchart for this study is provided in Fig 1.

### Development of a virtual-reality based gaze training tool

The initial phase of our project was dedicated to the development of the gaze training software. The aim was to create a tool that is easy to use, engaging, and that provides visual training

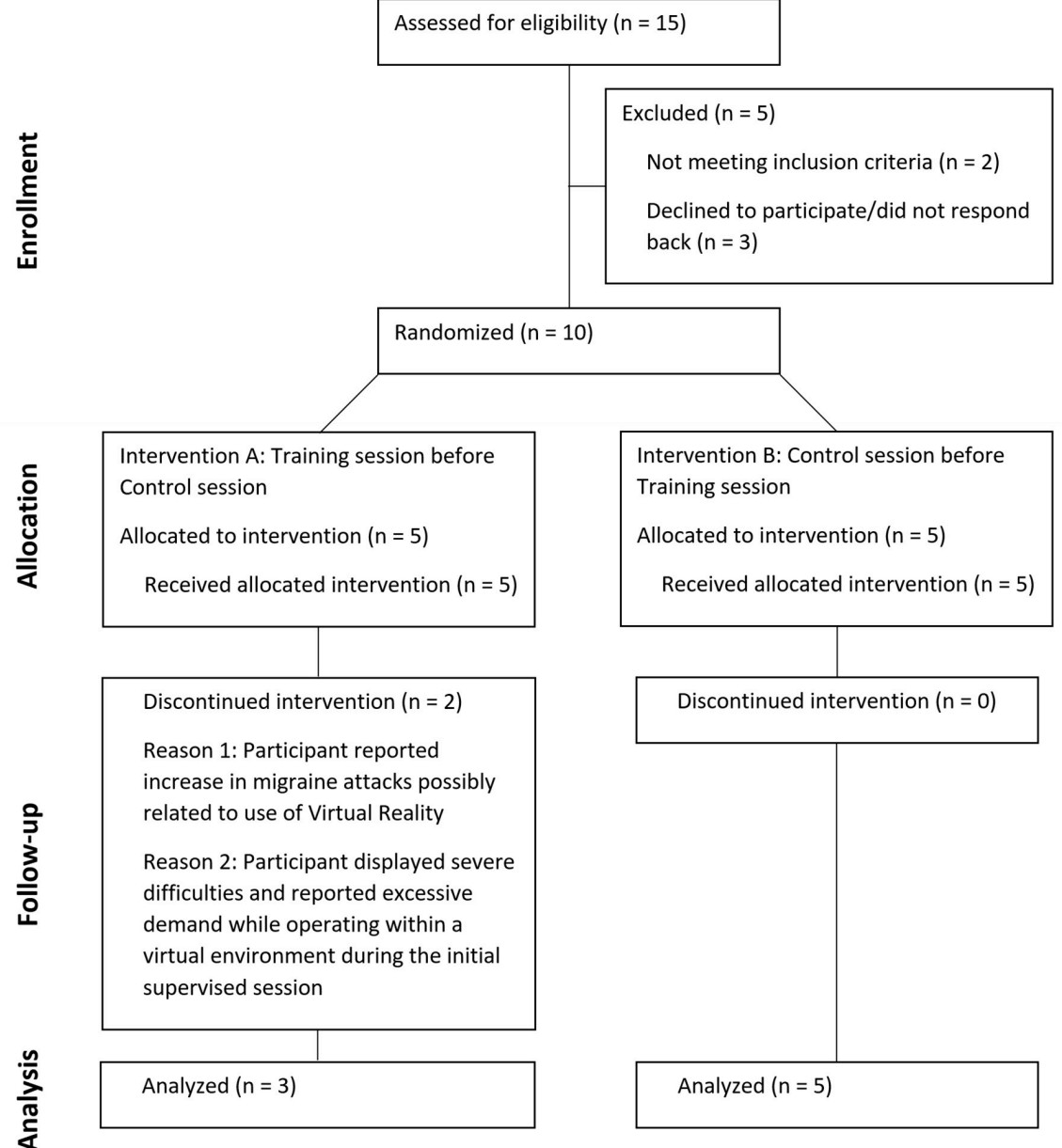

**Fig 1. CONSORT flowchart.** The CONSORT flowchart for the patient study described in this work.

tasks to motivate larger and more frequent eye movements. The training should be usable in an unsupervised at-home environment. The realizability and feasibility of this aim was demonstrated in studies such as by Ivanov et al. [14] and Kuyk et al. [17]. However, this unsupervised training approach mandated measurements that ensure risk-free execution of the training. Thus, training was designed such that no physical movement is required that would put the user or their environment at risk of accidents. All tasks were designed to be executed in seated or stationary standing position: While the viewing direction within the VR environment was controlled physically through head and body rotation, any form of locomotion was triggered solely through controller input. The implications of this sacrifice of real-world mobility—in a training specifically designed to improve the mobility of patients—will be further addressed in the discussion.

**Software and hardware specifications.** The training software was developed in the Unity3D game engine (Version 2021.3LTS), using the Pico XR SDK (version 1.2.4). The Pico Neo 2 Eye VR headset was used for development and training. It provides stand-alone functionality, meaning that no connection to a computer or any external tracking devices is necessary. The device features a 75 Hz display refresh rate and the VF per eye is stated to be 101˚ [30] according to the developer's specifications, though independent measurements have shown a VF per eye of 89˚ both horizontally and vertically [21]. The built-in eye tracker of the Pico Neo 2 Eye has a refresh rate of 90 Hz and an accuracy of 0.5˚ according to the device specifications, with an ideal eye-tracking range of 25˚ horizontally and 20˚ vertically [30]. The use of VR is possible while wearing glasses or contact lenses.

**Training tasks.** The training software consists of three visual tasks. Considering the unsupervised nature of the training, it wasn't practical to base the training on specific gaze instructions given to patients, as is typically done in supervised experimental training conditions [12, 19]. While patients could have received instructions before training, continuously monitoring patients over the course of the training to ensure that instructions are followed correctly would not have been possible. Acknowledging this, the training tasks were instead designed such that their success criteria naturally promote exploratory saccades and frequent eye movements, encouraging patients to develop own strategies and adaptive behavior. This follows the approach of previously mentioned studies by Nguyen et al. [15], Ivanov et al. [14], or Pambakian et al. [13]. In the following sections, the three visual tasks designed for the training are described:

- **Target tracking** In this task, a varying number of targets (starting at five) move across a two-dimensional area in front of the user in a random pattern (Fig 2A and 2D). To make the task more visually appealing and thematic, targets were displayed as cartoon-styled mice. At the start of training, the area's dimensions are 52˚ horizontally and 39˚ vertically, which roughly represents 30% of the visual angles of a healthy VF at 180˚×135˚. Two of the targets are marked at the start of the trial (visualized as a piece of cheese carried by the mouse, as illustrated in Fig 2A), and the user is asked to follow the marked targets with their gaze in order to not lose track of them. After 8–12 seconds, all targets stop their movements, and the marked targets change their appearance to become indistinguishable from the non-marked targets. At this point, the user is prompted to select the two formerly marked targets through input of the VR controller. Selected targets are revealed to be either correct or incorrect. A trial is considered to be successful if the user selects both correct targets and no or only one incorrect target. When selecting two incorrect targets, the trial fails.

- **Search Task** Inspired by visual search gaze training methods as applied in different previous studies [13, 14, 16], this task requires participants to search an area in front of them for specified visual cues. As in the Target Tracking Task, the default dimensions of this area are 52˚

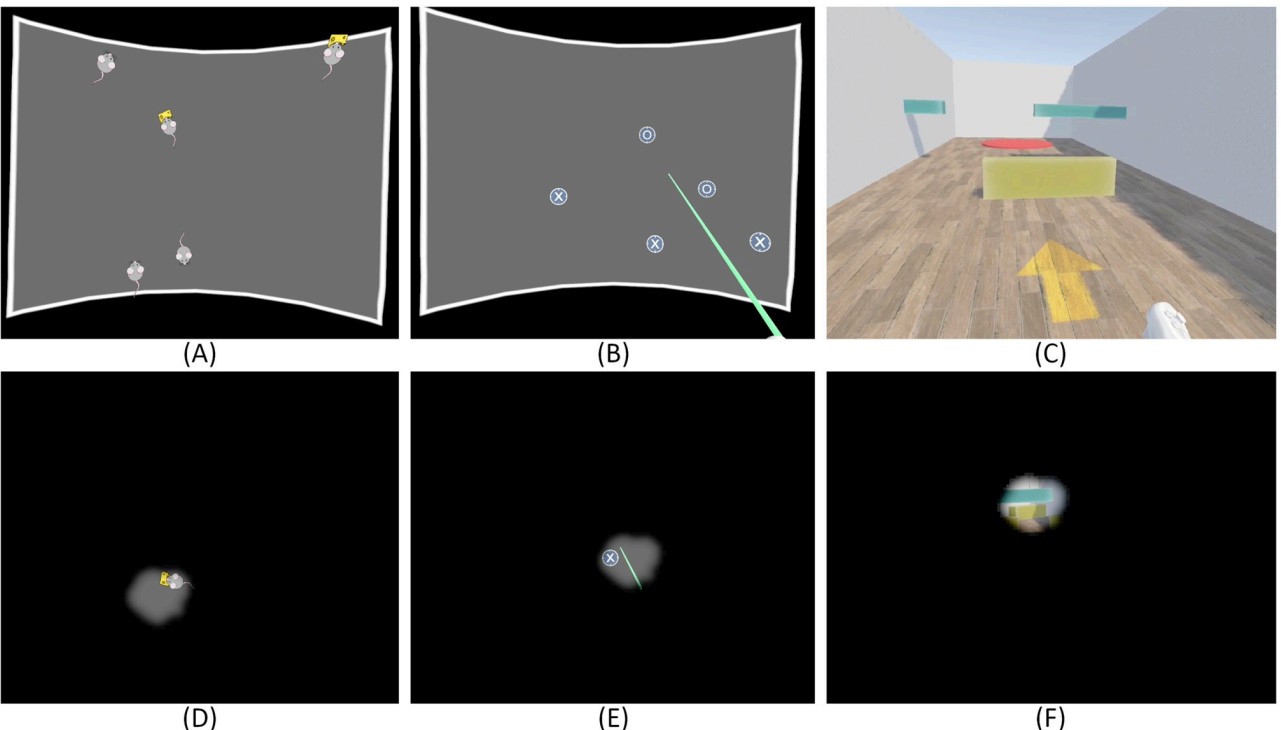

**Fig 2. Screen captures of the three visual tasks of the VR gaze training.** A: Target Tracking (marked targets are indicated by a piece of cheese); B: Search Task; C: Navigation Task; D: Target Tracking with simulated tunnel vision; E: Search Task with simulated tunnel vision; F: Navigation Task with simulated tunnel vision. The tunnel vision displayed in D-F has a 15˚ diameter. The tunnel vision simulation is added for visualization purposes in this manuscript only and was not present during participants' training. The grey training area has dimensions of 80˚×60˚ in these examples, representing an easy-to-medium difficulty level.

horizontally and 39˚ vertically. The objective of the task is to search for stationary targets, marked by a prominent cross symbol (Fig 2B and 2E), and to use the controllers to select as many of them as possible within a given time frame of 20 seconds. A total of three marked targets are placed in the defined area and once a target is selected, it is instantly moved to a new position inside the designated area, with a minimum distance of 30˚ from the previous position. This minimum distance is introduced to avoid targets re-appearing directly in the participant's VF, thus further promoting continuous scanning of the area to find additional targets. In addition to the targets marked with a cross, there are similar targets marked with a circle that serve as distractors to ensure that participants fully focus on a target before selecting it. Each trial was rated based on the number of marked targets that are found within the limited time frame.

- **Navigation Task** In the third task, participants are asked to navigate through a randomized obstacle course simulated in a virtual environment (Fig 2C and 2F). Using the controllers of the VR device, the participant is able to move at a dynamic pace, with a set maximum speed of $3\frac{m}{s}$ by default. The movement direction is controlled via the participant's body orientation, which is measured through the VR headset's orientation. The obstacle course is designed as a corridor with two left turn tiles, two right turn tiles and two straight tiles, each measuring 8 meters in both width and length (Fig 3). The six tiles are arranged in randomized order for a total of 90 unique layouts. Along the corridor, 12 randomized obstacles are placed. To motivate adaptive eye movements, obstacles have different height, shape and

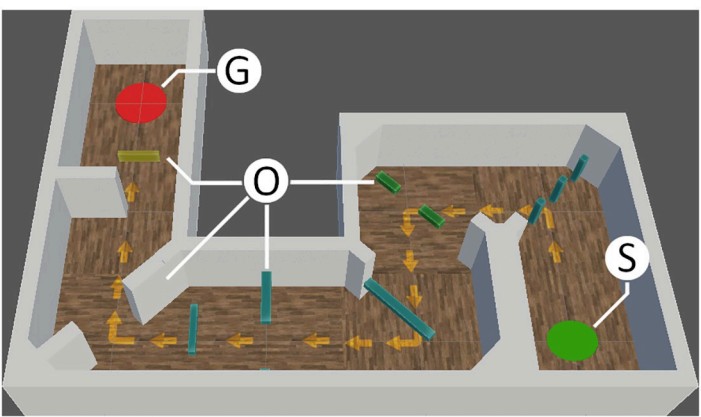

**Fig 3. Navigation task visualisation.** Top-down view on a randomized obstacle course of the Navigation Task. S: Starting position; O: Obstacles (example selection); G: Goal area.

movement patterns and can be divided into three categories: Near-ground obstacles require scanning of the lower visual area; Obstacles hanging at head-level require eye movements towards the upper regions of the visual area; Moving obstacles periodically move from one side of the walking corridor to the other and thus require dynamic and frequent eye movements to notice and avoid them. Collisions with obstacles or the walls bordering the walking corridor are indicated through a sound cue as well as a "bouncing" animation that moves the user's avatar back slightly. The goal of the obstacle course is marked with a prominent red circle, which the participant has to reach in order to successfully finish the trial. The trial was rated based on the duration required to move through the obstacle course, as well as the number of collisions during the trial. A trial was considered fully successful if the trial duration was below a specified threshold (default 60 seconds). A trial was considered completely failed if the goal was not reached within twice the duration of the threshold. Collisions reduced the threshold by approximately $\sim 15\%$ of its current value.

After each trial, participants are brought back to a selection menu in which they are able to inspect the rating and result of the trial as well as their overall progress, go back to the main menu or start the next trial. Additionally, participants had the option to mark the previous trial as "invalid", but were instructed to only mark trials as invalid if there were technical or external factors distorting the results. A video showcasing the three training tasks can be found in the S1 Video.

**Adaptive difficulty levels.** One of the design goals for the gaze training was to keep users engaged and motivated even throughout extended training phases. At the same time, the visual tasks should have low entry levels to make it easy for participants to get started and get used to the training tasks. To follow both premises, adaptive difficulty levels were introduced in all three visual training tasks. This means that the difficulty level of each individual task increases or decreases automatically based on the participant's current performance in that task, with the aim to ensure that the tasks remain at an appropriate level of difficulty to keep the user engaged and motivated. In the selection menu in-between trials, participants are informed about the difficulty level they have reached and about their progress towards the next difficulty level.

For the Target Tracking Task, higher difficulty levels translate to a larger bounding area in which the targets move, higher target movement speed, and a greater number of both marked

and unmarked targets. Similarly, increased difficulty in the Search Task leads to an expansion of the area in which targets are located, and increases the number of distractor targets while keeping the number of correct targets at three. In the Navigation Task, the maximum movement speed of the participant's avatar gradually increases with higher difficulty level, while at the same time reducing the time limit to move through the obstacle course without reducing the performance rating. Additionally, the speed of moving obstacles in the Navigation Task is adjusted to match the participant's increased movement speed, resulting in a faster-paced trial that demands quicker reaction times and heightened situational awareness to avoid obstacles.

**Suggested gaze pattern.** Preliminary studies [14, 31] suggest the potential of specific systematic eye movements—called gaze patterns—to positively impact gaze training. Following this, participants were encouraged to follow a suggested, systematic gaze pattern (visualized in Fig 4) while executing the training.

The shape of the gaze pattern was selected following the findings of a previous study [31] in which two popular gaze patterns were tested. One of the gaze patterns that were suggested to patients in this study (Fig 4) was found to lead to better results in both navigation as well as search tasks when compared to the competing pattern. However, findings of this and other gaze training studies [19] also suggest that training and application of specific, mandatory gaze patterns can potentially result in a reduction of the subjects' walking speed.

Thus, in this study, patients were given autonomy in choosing if—and to which degree—they want to follow the suggested gaze pattern. The pattern was visually introduced to the patients on a screen prior to the use of the gaze training, explaining its background and potential advantages.

In addition, after each training trial within the VR environment, participants received automated feedback in form of a "similarity value", a quantitative measurement of how closely their real gaze behavior matched the suggested gaze pattern. This quantitative measure of similarity between gaze behavior and suggested gaze pattern is evaluated at run-time using a Multimatch algorithm [32], and is described in detail in Appendix A in S1 Appendix.

## Experimental study

To test the influence of gaze training on the navigation performance in the real world, we designed an experimental randomized controlled crossover study. The layout consisted of two phases (Fig 5):

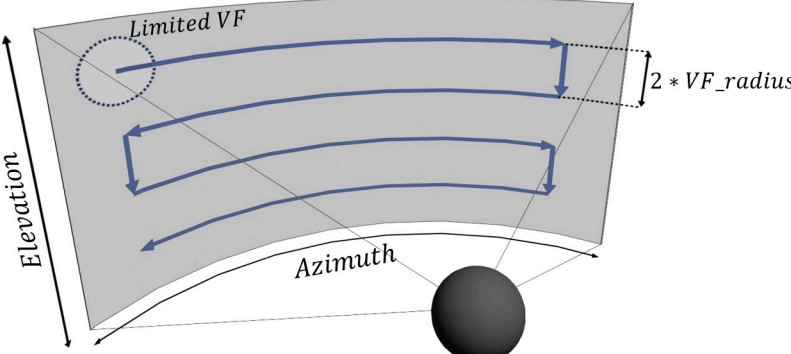

**Fig 4. Gaze pattern visualization.** Visualization of the pattern presented to the participants. The pattern starts in the upper left (or right) corner, moving along the azimuth axis to the opposite side. Then the pattern moves down along the elevation axis with an angular distance equal to the diameter of the participant's VF. This pattern is continued until the lowest area was scanned. Following this pattern ensures that a large visual area is covered by the limited VF in an efficient manner.

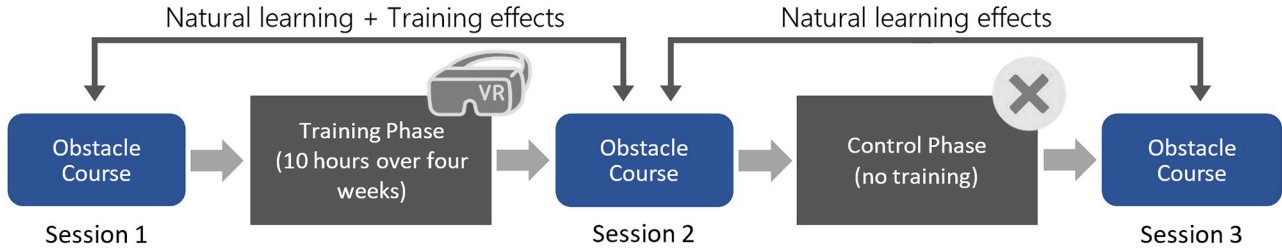

**Fig 5. Study structure.** Schematic of the study structure, including two phases of 3–4 weeks and the three in-person obstacle course sessions held. The order of training phase and control phase was randomized for each participant.

- In the training phase, participants used the developed VR gaze training software at home for 10 hours over the course of 3–4 weeks (20 training sessions, 30 minutes per day, 10 minutes per task).

- In the control phase, participants would follow their normal life routine over a similar duration without carrying out any gaze training.

The experimenter randomized the order of the two phases for each participant during scheduling using a random number generator, allocating five participants to the group starting with training and five participants to the group starting with the control phase. Before and after each phase, participants completed an in-person session in which their task was to move through a randomized real-world obstacle course (20 trials per session). Details about the setup and experimental environment for these sessions and the obstacle course trials can be found in Experimental setup. The purpose of the control phase was to account for any improvements that might occur during the task that are not correlated to the gaze training. Despite randomizing the obstacle course setup to minimize memorization effects, participants can still be expected to learn and improve their performance by becoming familiar with the base structure and types of obstacles in the obstacle course. This natural improvement in performance is considered the 'natural learning effect' of the experiment.

The impact of the training can thus be evaluated in three steps:

1. We assess the navigation performance in the real-world obstacle course before and after the training phase to determine the combined effect of both the potential training effect and natural learning effect. At this step, it is not yet possible to distinguish between the two effects.

2. Next, we assess the navigation performance before and after the control phase to find the natural learning effect displayed by participants, with no influence of gaze training.

3. By determining the differences between the "distilled" natural learning effect found in step 2 and the combined effect of training and natural learning found in step 1, it is possible to evaluate the effect that the developed gaze training has on real-world navigation performance. If the effect found after the training phase is significantly higher than the effect after the control phase, the gaze training can be considered successful.

**Ethics and clinical trial registry.** This study was proposed to and approved by the ethics committee of the Institutional Review Board of the Medical Faculty of the University of Tübingen (628/2018BO2) in accordance with the 2013 Helsinki Declaration. Patients were

recruited in the time from June 9, 2022 to November 22, 2022. All participants signed written informed consent forms. The study is registered as a clinical trial at the German Clinical Trials Register (DRKS) with the registry ID DRKS00032628. The registration was done retrospectively, as the study was originally considered as non-interventional observation study, not as clinical trial. Prompted by later feedback, this decision was reconsidered and the study was registered. The authors confirm that all ongoing and related trials for this intervention are registered.

**Study population.** 10 patients (one male, nine female), aged between 20 and 60 years (average 49.6 ± 13.0), participated in the study, two of which discontinued the study early on. Details about the reasons for discontinuation are provided in the Discussion. Participation criteria were the diagnosis with Retinitis pigmentosa, a VF between 5° and 30° diameter, a visual acuity of 0.1 or higher, and unrestricted mobility. It was tested in the first session that the participants are able to effortlessly see and recognize all targets and other visual features used in the gaze training, as well as all interfaces and menus required to operate the VR headset. The sample size was determined following the calculation for sample size in longitudinal studies comparing mean change with two time points, found in Rosner [33], equation 8.30), assuming a standard deviation of 20% (estimated based on Baroudi et al. [34]) of the mean navigation performance and an increase in performance of 25% after training, with an alpha of 0.05 and a power of 0.8.

Table 1 lists information about the eight participants that finished the study. The provided medical data is based on the most recent medical examination of each patient. The medical examinations only provided visual representations of the VF of patients, which are included in Appendix B in S1 Appendix. In addition to these, the VF was measured during the first in-person session using a VR based kinetic perimetry developed for this project. While these

**Table 1. Patient data.**

| Patient (Age / Sex) | Age of diagnose | Visual field (RE / LE) | Visual acuity[1] (RE / LE) | VF notes | Gaze training experience | VR experience | Vision correction |
|---|---|---|---|---|---|---|---|
| 1 (20f) | 14 | 7.62° / 8.26° | 0.40 / 0.40 | - | - | high | G/C[2] |
| 2 (57f) | 27 | 18.64° / 18.18° | 0.20 / 0.05 | spots[3] | VisioCoach[4] | - | G |
| 3 (55f) | 18 | 17.64° / 16.36° | 0.13 / 0.20 | - | - | - | G |
| 4 (47m) | 25 | 24.60° / 25.40° | 0.05 / 0.05[5] | - | - | low | G |
| 5 (59f) | 50 | 18.54° / 18.34° | 0.32 / 0.25 | spots[3] | VisioCoach[4] | - | G |
| 6 (59f) | 16 | 10.92° / 9.64° | 0.10 / 0.10 | - | - | - | G |
| 7 (40f) | 18 | 12.18° / 14.56° | 0.40 / 0.32 | spots[3] | VisioCoach[4] | - | G |
| 8 (60f) | 20 | 20.00° / 19.48° | 0.50 / 0.40 | - | - | - | G |

Summary of general patient data. In addition to the displayed data, all patients reported to have undergone Orientation & Mobility training with the white cane and were using the white cane as a navigation aid in everyday life. The column 'Visual field' reports on the average diameter of VF for right eye (RE) and left eye (LE) measured within the VR setup. Visual acuity reports on the Visual Acuity of patients measured during their most recent medical examination.

[1]Visual acuity is notated as decimal score.

[2]G = Glasses, C = Contact lenses. These refer to visual aids used in everyday life. Only contact lenses were worn during experimental trials, as glasses would interfere with the applied eye tracking device.

[3]The patient displays some spots of remaining vision in the peripheral field.

[4]A commercially available screen-based gaze training software for RP patients [35], applied in the previously mentioned studies by Ivanov et al. [14] and Roth et al. [16] and evaluated by Hazelton et al. [18].

[5]It should be noted that participant 4 does not meet the participation criterion of a visual acuity >0.1. This was discovered only after the start of training, since the initially provided medical examination report did not include results for the visual acuity. However, despite not meeting this criterion, the participant was still able to navigate the VR interface and did not exhibit any difficulties in recognizing the visual targets required for the tasks. Consequently, it was decided to continue with the study participation.

measurements do not have diagnostic validity, they provide a better estimation of the perceived visual area of patients within the VR setup. This approach also ensured consistency in the measurement of VFs between participants.

**Experimental setup.** The real-world obstacle course was set up in an area with the dimensions of 4.8 meters width and 9.0 meters length. Two static privacy screens (visible in Fig 6) were placed such that an S-shaped corridor is formed. This extended corridor had a length of 18 meters—assuming a pathway exactly along the middle of the corridor—and a width of 3 meters.

Within the path, different obstacles were placed in a semi-randomized layout. Each obstacle arrangement consisted of 12 large carton boxes measuring 120×60×60 centimeters. Six of the boxes were oriented horizontally, six vertically. The set of obstacles also included six low-height obstacles that required participants to step over them, measuring 120×20 centimeters. Lastly, three sheets of cloth of 60 centimeters width were hanging from the ceiling, their lower edge at a height of 150 to 170 centimeters, adjusted to be on participants' eye level. A total of 20 randomized obstacle layouts were created, with each layout being used exactly once per session. An example for one of these layouts is found in Fig 7. All obstacles were colored in blue to increase the contrast against the floor and background. The primary walking direction of the obstacle course was chosen such that participants were always facing away from the windows to avoid glare effects. The room was fully lit during all trials. A video showcasing the obstacle course in an example trial is found in the S2 Video.

During the trials, participants were wearing Pupil Labs Invisible eye tracking glasses [36]. The device provides a 0.5˚ accuracy and a 200Hz refresh rate [36] according to the technical specs provided by the seller. Tonsen et al. [37] find that the mean bias of gaze-estimation ranges from below 0.5˚ up to 2.5˚ based on the VF region, with mean sample errors of 5˚ to 6.5˚. Inertial measurements for tracking of head rotation use Madgwick's algorithm [38], but no specifics are given about their accuracy and precision. Timestamps for start and stop of each trial were measured and automatically stored using a custom smartphone application built with the Unity3D game engine.

**Experiment execution.** Each session was initiated with four unmeasured trials to familiarize the participant with the task, the types of existing obstacles and the shape of the walking corridor. After that, 20 measured trials were done. Details on the measured parameters within these trials are found in Measurement parameters.

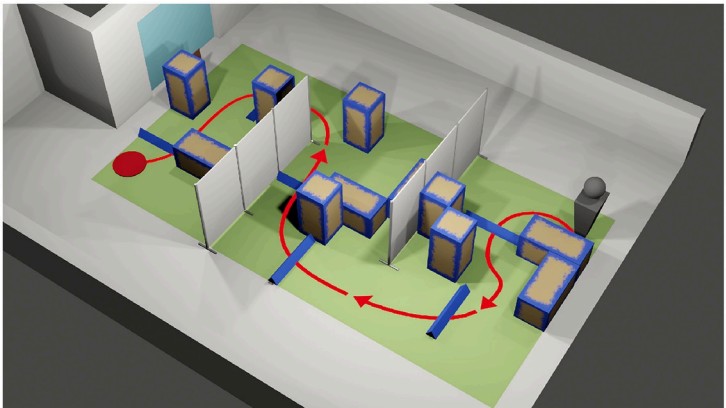 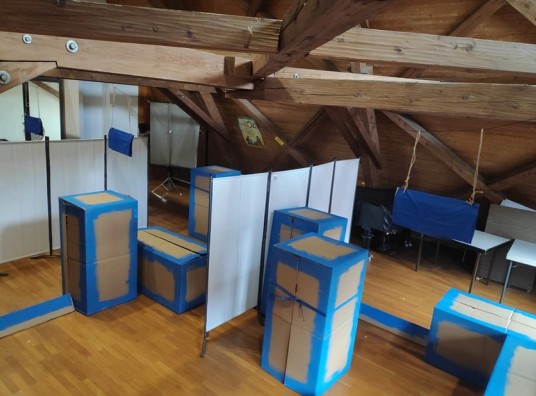

**Fig 6. Obstacle course layout.** Example of the real-world obstacle course in simulation (left) and actual setup (right). The simulated obstacle course is for presentation purposes only and was not used as part of the study.

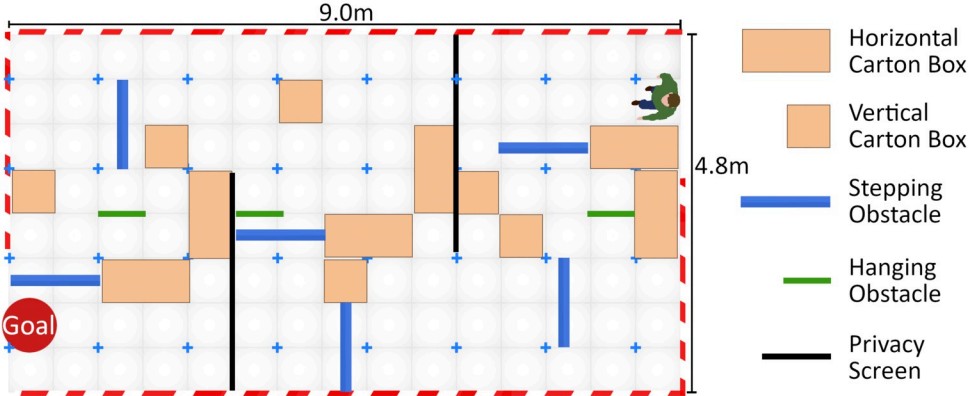

**Fig 7. Obstacle course schematic.** Example of one of the schematic layouts used to set up the obstacle course before each trial. Participants started each trial at the position that is marked by a person in the layout.

Prior to each trial, the experimenters positioned the obstacles according to one of 20 different layouts. Once the obstacle course was set up, the participant was guided to the starting position and directed to face forward towards the opposite side of the obstacle course. At this point, the Pupil Labs eye-tracking device was activated using the corresponding smartphone app. During this process, the participant had the opportunity to visually explore the scene, although only the initial third of the obstacle course was visible from the starting point due to the privacy screens. When everything was prepared and the participant confirmed their readiness, a second smartphone was used to start a timer and simultaneously play a sound cue, signaling the participant to start. While the participant navigated through the obstacle course, an experimenter followed them at a distance of approximately 3 meters, monitoring for any collisions that occurred during the trial. Upon reaching the designated goal area, the timer was stopped, accompanied by a second sound cue to indicate the completion of the trial.

During the session prior to the start of the gaze training, which could be either the first or second session, depending on the order of training and control phase, participants were introduced to the VR device and gaze training application. They were given the opportunity to familiarize themselves with the controls of the VR device and were guided through a tutorial of about 30 minutes duration in which the three visual tasks as well as the interface navigation within the gaze training application are explained. During this explanation, participants were also introduced to the suggested gaze pattern.

**Measurement parameters.** Two sets of measurement parameters were acquired as part of this study. The first set consists of the results of the real-world obstacle course trials, which were acquired during the in-person sessions. The second set consists of the performance and eye tracking information measured within the gaze training application during the four-week training phase. The following list summarizes all measurement parameters that were considered in the evaluation of this work:

**Real-world obstacle course measurements**

- **Trial duration** Trial duration in the real-world obstacle course trials describes the time required by the participant to move from the starting position to the designated goal area. It was measured with the custom smartphone application described in Experimental setup.

- **Collisions** This parameter describes the number of obstacle collisions that occurred during a trial. Collisions were visually observed and documented by an experimenter who was closely

following the participant during the trials. Collisions were categorized into two types: 'Full collisions' referred to frontal impacts where the obstacle was visibly and audibly struck—typically by the participant's foot -, requiring a complete adjustment or re-evaluation of the movement path. 'Light collisions' referred to situations where obstacles were grazed or lightly touched, without impeding the participant's motion. This distinction was made to acknowledge that patients may limit their efforts on the avoidance of collisions that would—in a real-life scenario—pose risks of accidents or injuries. Given that light collisions would usually not pose such risks, it is possible that patients put less priority on avoiding these types of collisions. Thus, if a shift from 'mostly full collisions' to 'mostly light collisions' is detected between two sessions, it can be considered a positive effect on navigation performance.

- **Dynamic visual field** Using Pupil Labs Invisible eye tracking glasses, both the direction of gaze relative to the head and the orientation of the head itself are measured during the real-world obstacle course trials. Based on these two parameters, the dynamic visual field (DVF) is calculated. We define the DVF as the visual area observed over a specified amount of time. When fixating a single point, a person with tunnel vision would only be able to observe the area within their normal VF—in this context, this could be described as the 'static' visual field. However, as soon as the person starts moving their gaze, they will automatically explore and observe new areas of their visual surroundings. Measuring this observed area over a fixed duration results in the DVF. Notably, the DVF only increases if new visual area is explored that has not already been observed within the specified time frame. In this work, the time frame for the DVF is set to three seconds. This means that the DVF at any point of time is defined as the observed visual area over the last three seconds. Averaging the DVF for all measured samples within a trial provides the average DVF for that trial. The DVF is reported and evaluated as a percentage change, showing how much the DVF increased or decreased over the course of the training or control phase. An increase of average DVF of 10% indicates that the person was able to observe 10% more of their surroundings. DVF is calculated considering all eye tracking samples, both during fixations and during saccades, since detection of potential obstacles or points of interest is possible even during eye movements [39]. A detailed description for how the DVF is calculated and computed is found in Appendix B in S1 Appendix.

- **Saccade characteristics and gaze pattern similarity** In addition to the DVF, the eye tracking data were also used to determine saccades during the trial, following an approach by Nyström et al. [40]. In a first step, the current gaze speed for each sample frame of the captured eye tracking data was calculated, summing both eye movements and head rotation angles. The data was smoothed out by applying a moving window average over five frames ($\sim$ 25ms). Next, the mean $\mu$ and standard deviation $\sigma$ of the noise of the eye tracking samples were determined. For this, only samples with angular gaze speed below 100°/s were considered. Following the approach of Nyström et al., a saccade was detected when a peak in the gaze speed surpassed a threshold $v_{max} = \mu + 6 * \sigma$. If a saccade was detected like that, the start- and end point of the saccade were determined based on a second threshold $v_{onset} = \mu + 3 * \sigma$. Appendix C in S1 Appendix shows an example graphic that visualizes saccade detection using this approach. The saccades were analyzed for different characteristics. The first characteristic was the ratio of exploratory saccades, defined as the number of saccades larger than the average visual field radius divided by the total number of saccades per trial. Next, the saccade frequency was determined as the average number of saccades per second. The average ratio between the horizontal and vertical components of saccades was determined to assess whether patients move their gaze more vertically or horizontally. Following an approach by

David et al. [41], we analyzed the direction of each saccade relative to its preceding saccade. This provides additional insight into the patients' gaze behavior. Lastly, the saccades were used to calculate the gaze pattern similarity value as described in Suggested gaze pattern. Saccades were determined by considering world-centric gaze, meaning that both the direction of the eyes as well as the rotation of the head were considered.

**Gaze training measurements**.

- **Target tracking task performance** The performance of the Target Tracking Task—the task in which participants had to track and select a number of marked targets—was evaluated based on the number of incorrectly selected targets per trial. Each trial's performance was measured on a point scale, where trials with no incorrect targets selected were rated with two points, and trials with one incorrect target equated to one point. However, due to the gradual change in difficulty levels of the task—described in Training tasks—rating each trial in the same way would not result in a good approximation of a participant's total performance, as higher difficulty levels are likely to result in lower success rates. For the Target Tracking Task, the main factor that influences the difficulty of a task is the number of marked targets that must be tracked simultaneously. It is not feasible to compare a trial with only two marked targets to a trial with three or even four marked targets. Thus, to achieve a balanced approximation of task performance, only trials with four marked targets were considered, which is 45.1% of all measured trials.

- **Search task performance** In the Search Task, the base performance can simply be measured as the number of targets found and selected during the fixed 20-second time period of a trial. However, this again does not consider the change in difficulty level, which results in a larger or smaller area that has to be scanned to find the targets. To account for this, the Search Task performance score was adjusted based on the size of the search area in which targets would be placed, such that $P_{adj} = n * (w_{area} * h_{area})$, where $P_{adj}$ is the adjusted performance score, $n$ is the number of targets found and $w_{area}$ and $h_{area}$ being width and height (in visual angles) of the search area. In other words, to achieve the same performance score in a search area four times larger, the participant would have to find four times fewer targets.

- **Navigation task performance** The performance of the Navigation Task consists of two measurement parameters, both of which are reported on separately. The first parameter is the trial duration, which is the time taken from start to finish of the navigation course. The second parameter is the number of collisions during a trial. The layout of the obstacle course did not change with varying difficulty levels, making trials of different levels of difficulty more comparable to each other than in the other two tasks.

- **Gaze direction and dynamic visual field** Similar to the real-world course trials, both head-centered gaze direction and head rotation were measured in all three visual tasks of the gaze training, using the VR headset's built-in Tobii eye tracking device. This data was used to calculate the DVF on a frame-by-frame basis, using the combined gaze direction.

It must be noted that the performance scores calculated for the visual tasks of the gaze training are just an approximation of a participant's actual skill level at different stages of the training, and are influenced by the methods that are applied to consider and eliminate the impact of varying difficulty levels on the performance.

**Questionnaire.**   Five times during the training phase—following the initial training session and subsequently after every five training sessions -, participants were requested to complete a questionnaire to assess subjective ratings of enjoyment, motivation, stress, eye strain and other related factors. The questionnaire always featured the same questions, with seven of

the questions following a 10-point Likert scale format and four questions allowing for free-form answers:

**Questions to rank from 1 to 10:**

- **Enjoyment**—To what extent do you find each of the visual tasks enjoyable?

- **Motivation**—How motivated are you to improve your performance in each of the visual tasks?

- **Easiness**—How would you rate the ease of carrying out each of the visual tasks?

- **Stress**—To what degree do you experience stress while executing each of the visual tasks?

- **Eye Strain**—How straining is each visual task on your eyes?

- **Intuitiveness**—How intuitive is the use of the gaze training software?

- **Discomfort**—How much physical discomfort do you experience while wearing the VR headset?

**Questions with free-form answers:**

- **Feedback for gaze training**—Which aspects of the gaze training application did you perceive as especially positive or negative?

- **Feedback for VR device**—Which aspects of the Virtual Reality headset did you perceive as especially positive or negative?

- **Feedback and suggestions**—What changes or improvements would you like to see implemented in the gaze training application?

- **Technical issues**—Did you encounter any technical issues during the training? If so, please describe.

### Evaluation process and statistical methods

In the real-world obstacle course tasks, effects for the four measurement parameters were tested. Each parameter was tested using two different paradigms: First, the effects of training- and control phase were assessed individually by testing the data acquired in the session before the respective phase against data acquired in the session after the phase (Pre-Post test). Second, the two effect sizes from training and control phase are tested directly against each other to determine if training effects significantly surpass effects of the control phase (Training-Control test). For this test, delta values for each trial are calculated: For example, the difference between the first trial of the session before the training/control phase and the first trial of the session after the training/control phase is calculated. This way, the effects for each phase can be expressed as a set of delta values. By testing the set of delta values from the training phase against the delta values from the control phase, statistical significance between effect sizes can be evaluated. The following sections describe the statistical models and pre-processing steps for each measurement parameter.

- **Trial duration** For the Pre-Post test, a Linear Mixed Model (LMM) with trial duration as dependent variable is used. As a fixed factor, the 'pre-post condition' is applied. This binary parameter signifies whether a respective trial originates from the real-world session before or after the relevant phase. In addition to the fixed effect, participants were included as a random factor in the model, considering both random intercept (to consider different innate

skills) and random slope (to consider different learning rates). Since trial duration results were not normally distributed and instead followed a right-skewed distribution, a logarithmic transformation was applied to the data to better meet the requirements of an LMM. A QQ-plot for the results with logarithm taken is found in (Appendix D in S1 Appendix). The Training-Control test was tested mostly analogous to the Pre-Post test, again using an LMM. Delta trial duration was used as dependent variable, with the respective phase (training or control) as fixed factor.

- **Collisions** The collision parameter does not meet the requirements of a standard LMM, as its values are discrete count data, rather than continuous and normally-distributed. In addition, data was highly zero-inflated, with more than half of all trials (64.0%) showing zero collisions. Thus, for the Pre-Post test, we applied a negative binomial Generalized Linear Mixed Model (nbGLMM) which is suited for this type of data [42, 43]. As before, the pre-post condition was considered as a fixed effect and participants were considered as a random effect, with one model testing the effects over the training phase and a second model testing the effects over the control phase. For the Training-Control test, values were no longer zero-inflated, as the delta values could be both positive and negative. Thus, a Generalized Linear Mixed Model (GLMM) was applied.

- **Dynamic visual field** The DVF was found to follow normal distribution quite well (the QQ-plot is found in Appendix D in S1 Appendix) and data is continuous, allowing the use of an LMM for both Pre-Post test and Training-Control test with no additional transformations. Analogous to the other measurement parameters, the Pre-Post test uses absolute DVF values as dependent variable and the pre-post condition as fixed factor, whereas the Training-Control test uses delta values of the DVF as dependent variable and the respective phase as fixed effect.

- **Gaze pattern similarity** The different saccade characteristics—ratio of exploratory saccades, saccade frequency, ratio of vertical to horizontal gaze movements, as well as the change in directions of saccades, were evaluated analogous to the DVF. The same is true for the gaze pattern similarity.

While the order of the obstacle trials was changed between sessions, each of the 20 obstacle trial layouts was used exactly once per session. This ensures that the effect of different layouts on the performance within the obstacle course does not have to be considered in the statistical models.

Regarding the results of the Virtual Reality gaze training, it was modeled and analyzed how the task performance as well as the DVF in all three visual tasks changes over the course of the training.

- **Target Tracking Performance** As described in Measurement parameters, the performance in the Target Tracking Task is based on the number of incorrectly selected targets at the end of a trial. To measure this, a point scoring system was employed, where a score of 2 points was assigned for trials with zero errors, 1 point for trials with one error, and 0 points for trials with two or more errors. This means that the Tracking Task Performance can be treated as count data, and thus a Generalized Linear Mixed Model (GLMM) was employed for the analysis. Fixed factors of the model are the training session number (from 1 to 20) as well as the number of the current trial within the training session, as both can be assumed to have an influence on the task performance. Once again, participants were considered as random factor with both random intercept and random slope.

- **Search Task Performance** Search Task Performance was measured as the number of stationary targets found in a 20 second interval. QQ-plots found it to roughly follow normal distribution, making the use of an LMM suitable for analysis. As before, fixed factors of the model included training session and trial number, participants are considered as random factor.

- **Navigation Trial Duration** Similar to the real-world obstacle course trials, the trial duration of the Navigation Task trials was found to be right-skewed, thus the logarithm was taken for the analysis. An LMM was employed analogues to the previous analysis of the Search Task Performance.

- **Navigation Trial Collisions** The number of collisions per trial can be treated as zero-inflated count data, similar to the collisions in the real-world obstacle course. This indicates the need for a nbGLMM, where training session and trial number are treated as fixed factors, participants as random factor.

- **Dynamic visual field** The DVF was analyzed the same for all three visual tasks using an LMM. No transformation was required, as DVF followed normal distribution in all three tasks according to QQ-plots. Following the previous models, the DVF was tested against the training session and trial number as fixed factors, with participants being considered as random factor.

The alpha level that determines the threshold for statistical significance was chosen as 0.05 for all models. All errors are reported as the standard deviation of results. Analysis was done using R and the RStudio graphical interface with the nlme and lme4 library. The detailed models and results of the statistical analyses can be found in Appendix D in S1 Appendix. The results of the questionnaires are reported on qualitatively, as the number of samples is too low for statistical analysis.

For the real-world obstacle course results, it must be mentioned that 61 out of 480 measured trials did not include complete gaze-tracking data (25 of these trials included head-centric gaze data but no head rotations, 36 trials were missing both eye- and head-tracking data) and thus had to be discarded from the analysis of DVF. This loss of data was likely caused by a shaky contact of the eye tracking device and was only detected late in the experiment phase. The data loss affected three sessions in particular: The eye-tracking data was lost or incomplete in 15 out of 20 trials in the second session of participant 4, 18 out of 20 trials in the second session of participant 6, and all 20 trials in the first session of participant 1. Thus, for participant 1, no results for the change of DVF over the control phase could be evaluated.

## Deviations from original study protocol

This chapter lists the deviations of the actual methods from the original study protocol:

- The initial proposal outlined the use of a Fove-0 VR headset for the study. In the time between the ethics application and the start of the study, new VR devices became available that were better suited for at-home training, notably the Pico Neo 2 Eye, which ultimately became the chosen hardware for this study. The main advantage of the Pico Neo 2 Eye compared to the Fove-0 is its self-contained hardware, as it does not require any external hardware setups.

- The protocol allocated a total of 30 RP patients for the study, divided into two groups of 15 patients each: A training group and a control group. During the patient acquisition phase it became clear that the number of RP patients interested in study participation would not

allow for this study population size. Thus, the design was changed to a crossover study, with all patients carrying out both training and control phase in randomized order, as is described in Experimental study. Additionally, the protocol provided for inclusion of an additional group of visually healthy participants as control. This plan was discarded because the low relevance of the results to be obtained from this group would not have justified the additional time and material effort.

- The setup of the real-world obstacle course used in the study deviates from the one described in the protocol. The protocol outlined a 68m long and 1.3m wide corridor. At the time of the study, no location was available that would have allowed for a course of these dimensions. Thus, the course was adjusted to the dimensions described in Experimental setup.

- The protocol planned for a mandatory 'eye motion' task as part of the gaze training. As was mentioned in Suggested gaze pattern and will be further addressed in Discussion, the part was instead included as a voluntary task, following the findings of a preliminary study focused on the effects of gaze patterns in gaze training [31].

- In addition to the real-world obstacle course, the protocol provided for a performance evaluation in a realistic city environment within the virtual world. This plan was discarded because the development of a realistic 3D city environment would have been beyond the available time and expertise for the study.

- Lastly, the methods for statistical analysis were changed. The original protocol outlined the use of t-tests and mixed model repeated measures ANOVA analysis. Over the course of the study assessment, it was decided that Linear Mixed Models as well as negative binomial Linear Mixed Models are more suitable for the statistical evaluation of the acquired data.

## Results

### Real-world obstacle course

A total of 480 real-world obstacle course trials were absolved, split among 8 participants with three sessions each. The raw result tables for the trials are found in the supplementary material of this work. Fig 8 shows the results for navigation performances and the DVF of the eight participants.

**Navigation performance: Trial duration and number of collisions.** After the training phase, participants displayed a significant improvement in trial duration by 17.0% compared to the performance before the training ($p < 0.001$), decreasing the average trial duration from 37.2 ($\pm$12.3) seconds to 30.9 ($\pm$8.68) seconds. The average number of collisions per trial decreased by 50.0% after training ($p < 0.001$), from 0.513 collisions per trial to 0.256 collisions per trial. A comparison with the results before and after the control phase shows that the training phase was significantly more effective in improving the average trial duration ($p < 0.001$) and reducing the number of collisions ($p = 0.0165$) than the control phase. The average trial duration had improved by 5.9% after the control phase, from 34.8 ($\pm$12.7) seconds to 32.7 ($\pm$9.87) seconds. The average number of collisions per trial improved by 10.4% after the control phase, from 0.391 to 0.350 collisions per trial. Overall, the results suggest that the training phase was significantly more effective in improving navigation performance compared to the control phase.

Out of the four obstacle types in the real-world obstacle course—horizontal box, vertical box, stepping obstacle and hanging obstacle—the type that caused most collisions is the stepping obstacle with a total of 76 full collisions and 53 light collisions in all 480 trials. However, it

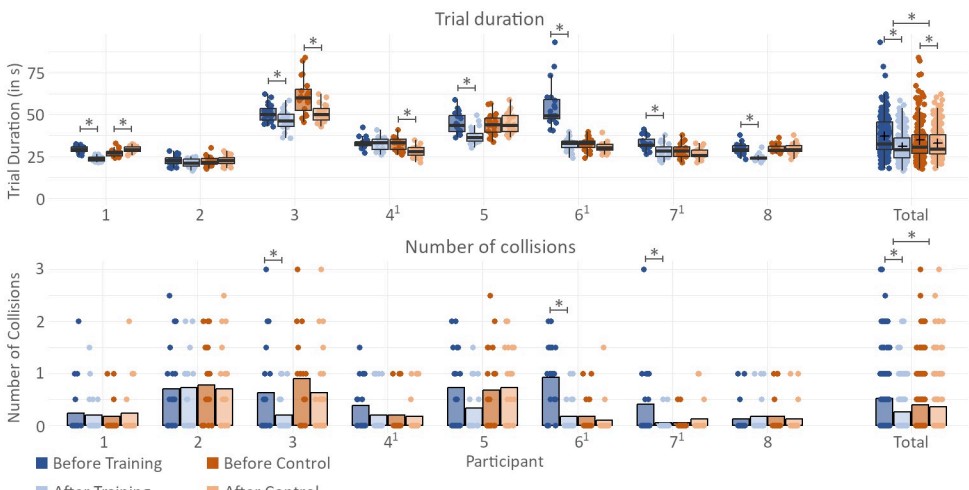

**Fig 8. Obstacle course results.** Participants' performance in the real-world obstacle course trials before and after each of the two phases. *indicates significance ($p < 0.05$). P-values for individual patients were evaluated by applying the statistical models to a subset of the data containing only trials of the respective patient. [1]marks participants who carried out the training phase before the control phase; all other participants started with the control phase and carried out the training phase afterward.

is shortly followed by the hanging obstacle at 75 full collisions and 52 light collisions. Considering that each obstacle course layout features six stepping obstacles, but only three hanging obstacles, it can be stated that the hanging obstacles pose the highest risk for collisions. Very few collisions were tracked for the horizontal and vertical boxes, with 6 full collisions and 8 light collisions for horizontal boxes and 3 full collisions and 13 light collisions for vertical boxes.

**Visual performance: Dynamic visual field and gaze patterns.** Using the eye-tracking data collected during the real-world obstacle course trials, it is possible to evaluate how the average DVF of participants—the visual area observed over time—changed over the course of training and control phase, as shown in Fig 9. Although the average increase in world-centric DVF of 4.41% is found to be significant ($p < 0.001$) when evaluating the data before and after training, the effect is not significantly larger than the increase in DVF displayed after control ($p = 0.394$) at 2.06%. Three of eight participants (1,3,6) display a notable increase in world-centric DVF after the training phase, with two participants (4,8) showing decreases. When considering only head-centric eye movements, no significant change in DVF is found for the average DVF after either phase, with a change of -0.052% ($p = 0.175$) after training phase and 0.108% ($p = 0.383$) after control phase and no significant effect between training and control ($p = 0.148$). While this suggest that the increase in DVF originates mainly from a change in head movements, rather than eye movements, two of the three participants (3, 6) with notable increase in world-centric DVF also show similar increase in gaze-centric DVF.

The results of the evaluation of the gaze pattern similarity—the value describing how closely the participants' displayed gaze movements match the suggested gaze pattern that was presented in Fig 4—is shown in Fig 10. There is no significant increase found after either of the two phases ($p = 0.168$ for training phase, $p = 0.147$ for control phase), and the similarity values give no indication that participants were actively following the suggested gaze pattern.

Similar to the results of DVF and gaze pattern, a direct comparison between effects of training and control phase on different saccade characteristics does not show significance. Saccade

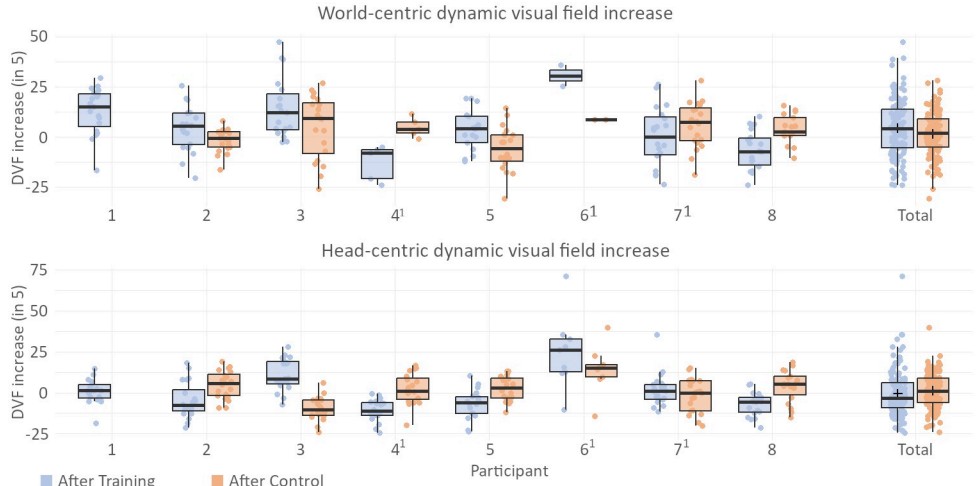

**Fig 9. Dynamic visual field.** Increase of the dynamic visual field in the real-world task after training phase or control phase respectively. Top graph shows the results based on world-centric gaze data (considers both head- and eye movements) whereas the bottom graph shows results based on head-centric gaze data (only eye movements, no head rotation considered). DVF was calculated over a 3 second rolling window. [1]denotes participants that carried out the training phase before the control phase.

frequency was found to increase by 3.20% after training and decrease by 1.73% after the control phase ($p = 0.36$). Exploratory saccade ratio decreased by 1.32% after training and increased by 0.69% after control ($p = 0.44$). 6.5% more vertical eye movements were displayed after training, 0.39% more after the control phase ($p = 0.09$). No significant changes are found in the direction of saccades relative to the preceding saccade ($p = 0.068$ for angles of 45˚-135˚, $p = 0.53$ for angles of at least 135˚). It can be noted that a strong variation between individual patients was found for the results, which is further elaborated in the Discussion.

## Virtual-reality gaze training

For the gaze training, a total of 3125 Target Tracking trials, 3205 Search Task trials, and 2583 Navigation trials were evaluated, distributed across the eight participants and ∼20 training

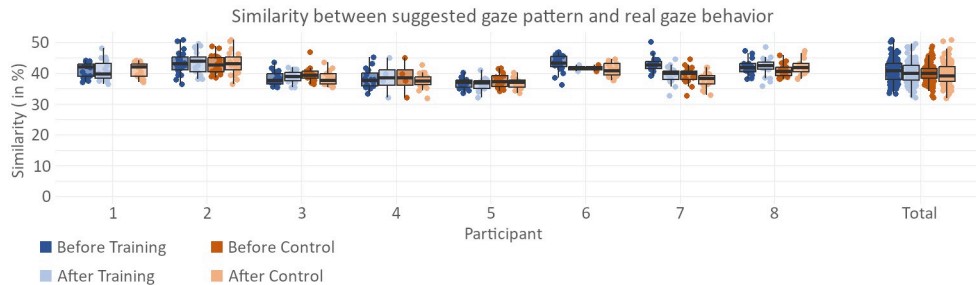

**Fig 10. Gaze pattern results.** The figure shows the similarity between the gaze pattern displayed by participants and the suggested systematic gaze pattern described in Suggested gaze pattern. Values between 0.3 and 0.5 are common for naive gaze behavior, whereas values between 0.6 and 0.8 can be expected when a subject is actively following the gaze pattern.

sessions per participant. Fig 11 shows the visual task performance and DVF within the virtual environment over the course of the 20 training sessions.

Most participants improved both in performance and in average DVF in all three tasks. In the Search Task, the average task performance increased by 35.5% ($p < 0.001$). In the Target Tracking Task, performance increased by 13.9% ($p < 0.0032$). The average trial duration in the Navigation Task decreases by 61.0% ($p < 0.001$), the number of collisions is reduced by 80.3% ($p < 0.001$). Despite showing the lowest task performance increase, the Target Tracking Task evoked the highest increase in average DVF over the course of training, with an increase of 43.4% from beginning to end of training ($p < 0.001$). The Search Task followed with an increase in DVF of 29.9% ($p < 0.001$), and the Navigation Task resulted in the lowest average DVF increase out of the three tasks at 19.8% ($p < 0.001$).

**Questionnaire.** The questionnaires filled by participants over the course of the gaze training give insight into qualitative results. Fig 12 shows the average results of all five questionnaires for the seven ranking questions related to the VR gaze training shown in Questionnaire.

Overall, the Search Task was rated most positively, with high enjoyment and motivation for improvement, as well as low perceived stress and low eye strain reported. On the opposite side, the Target Tracking Task was rated most negatively, with the lowest task enjoyment and motivation to improve upon previous results, and highest stress and eye strain reported out of the three tasks. The questionnaire ratings show high standard deviation between participants, with scores oftentimes ranging from 1 to 10 within the same conditions. Some of the scores of individual patients do not align with verbal feedback given after the study and may thus be a result of misinterpretation of the question. Still, these consistently high standard deviations indicate that the different aspects of the training tasks, such as motivation, perceived difficulty, and enjoyment, are highly subjective. In addition to the score ranking, participants also gave general feedback for the gaze training, both in the questionnaires and in the in-person training sessions. Some participants stated minor technical issues both with the VR headset and the developed software. Additional suggestions included more variety in tasks or task

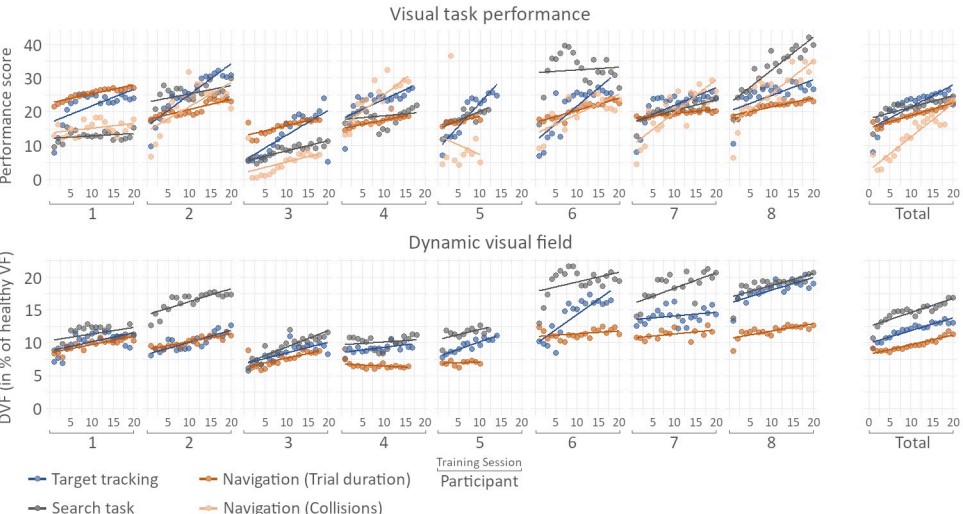

**Fig 11. Gaze training results.** Task performance score (top) and DVF (bottom) of the three visual training tasks. The Performance score for trial duration was calculated based on the logarithm of the trial duration. The reported DVF is world-centric, measured over a 3 second rolling window and given as percentage of a 180˚x135˚ area (approximately the dimensions of a healthy VF).

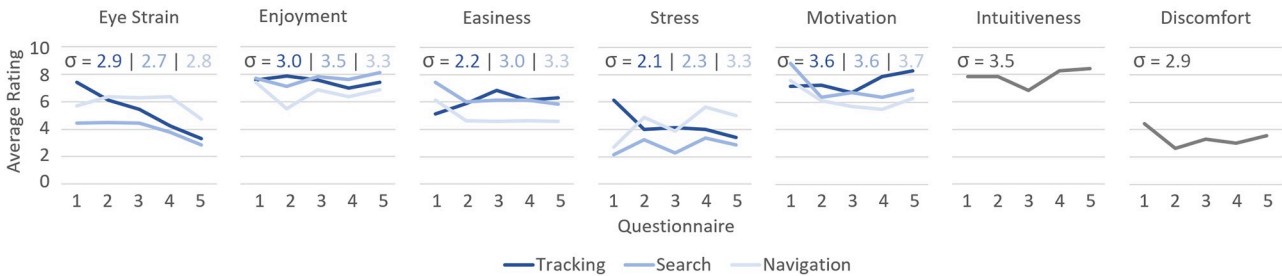

**Fig 12. Questionnaire results.** Average rating of different aspects of the gaze training. Questionnaires 1–5 were filled after 1, 5, 10, 15 and 20 training sessions. A rating of 10 equals "very high", a rating of 1 equals "very low". The $\sigma$-values indicate the average standard deviation over all five questionnaires of the respective task.

visualization, such as adding different backgrounds; As well as quality-of-life changes, such as more fluent turn animations for the moving targets in the Target Tracking Task, or a better indication before the application switches to the selection menu at the end of a trial.

The raw data results for the obstacle course trials, the VR gaze training, as well as the questionnaires are found in the S1 File.

## Discussion

In this study, the effects of 10 hours of training with a VR based gaze training on the navigation performance in a real-world obstacle course task was evaluated in eight Retinitis pigmentosa patients. It is found that the navigation performance increase over the training phase significantly surpasses the natural learning effect found after the control phase, suggesting that Virtual-Reality based gaze training can improve navigation performance of patients with Retinitis pigmentosa. While notable changes in DVF are found in individual patients, the group's average DVF increase after training was not found to significantly surpass the effect of the control phase.

Analysis of the real-world obstacle course results for each individual participant reveals a significant variability in the impact of gaze training on the DVF. Among the eight participants whose results were evaluated, three demonstrated a prominent increase in DVF during the real-world task after training (Fig 9). Notably, two of these participants (participant 3 and 6) also exhibited significant improvements in both trial duration and collision avoidance (Fig 8), with the third participant (participant 1) showing a significant improvement only in trial duration, with no corresponding increase in collision avoidance. These findings imply a positive correlation between an expanded DVF and improved navigation performance. However, it is worth noting that improvements in trial duration and collision avoidance were also observed in participants who did not experience any changes in DVF following training. Participants 5 and 7 exhibited significant improvements in trial duration and noticeable improvements in collision avoidance, despite no discernible change in DVF. Likewise, participants 4 and 8—the only participants to display a decrease in DVF after training—did not exhibit any negative effect on trial duration or collision avoidance, which would be expected when assuming a direct correlation between DVF and navigation performance.

In summary, it is possible that an increased DVF has a positive effect on navigation performance, indicated by the fact that all three patients displaying such increase in DVF also display improvements in navigation performance. However, the opposite statement cannot be made: A lack of increase in DVF—or even a decrease in DVF—in a patient does not imply the absence of improvements in performance, showing that improved trial duration and collision

avoidance are not a direct indicator for changed gaze behavior. This further suggests the presence of other factors through which gaze training influences navigation performance. Participants 4 and 5, for instance, both demonstrated improvements in the Navigation Task over the course of the training despite minimal changes in their DVF (Fig 11). This aligns with the results obtained from the real-world obstacle course (Figs 8 and 9), where participants also displayed improved navigation performance with no increase or even with decrease in DVF. Based on these findings, it is plausible to speculate that the Navigation Task facilitated the development of spatial awareness in these and maybe other participants. It is also possible that through the Navigation Task, participants adapted to previously less familiar types of obstacles, such as obstacles hanging from the ceiling.

In Suggested gaze pattern, a preliminary study was mentioned [31]. Here, we assessed whether systematic gaze patterns could assist people with tunnel vision condition in navigation and obstacle avoidance. For that, a gaze-contingent simulation of a 20˚ diameter tunnel vision condition was employed on visually healthy participants within a Virtual Reality environment, and the participants navigated through a virtual obstacle course while following different gaze patterns in a supervised experimental setup. One of the tested gaze patterns—a mostly horizontal, serpentine scanning motion—was found to significantly reduce obstacle collisions by 32.9% and increase the DVF by 8.9% compared to trials without systematic gaze movements. This came at the cost of a significantly lower movement speed, resulting in an average of 24.6% longer trial duration when following the gaze pattern. The study concluded that the gaze pattern has the potential to enhance visual performance in the presence of tunnel vision and suggests that introducing the gaze pattern in gaze training for individuals with tunnel vision could have beneficial effects. This is in line with a study by Nelles et al. [12], in which hemianopia patients were introduced to a similar gaze pattern in supervised training, displaying significant improvements in visual search after training. However, it is important to note that our preliminary gaze pattern study did not involve any real RP patients and no effect of the execution of gaze patterns in a real-world setting were evaluated in this study. Given the non-conclusive nature of the results of this previous study, the execution of the suggested gaze pattern was included as a voluntary task for the gaze training presented in this work.

As the evaluation of the gaze pattern similarity before and after the training phase indicates, the participants have not adapted this gaze pattern. This is also reflected in participants' statements after the training. Six out of the eight participants (1, 2, 3, 4, 5, 8) reported that they did not follow any specific gaze movement strategies. The predominant reason stated was that they "didn't think of it" during the trials. Participants 6 and 7 reported to follow individual gaze patterns. Participant 6 described a radial gaze pattern, moving the gaze in a small circle at first and then in a second, larger circle. She reported to have adapted this gaze pattern strategy during the Target Tracking Task of the gaze training and is now using it successfully even in everyday life. Participant 7 reported to follow a cross-like gaze pattern, moving the gaze vertically from bottom to top, then left to right. However, she reported to have adapted this gaze pattern only for the real-world obstacle course trials and not during the gaze training. Overall, the outcome implies that systematic gaze patterns are not easily and voluntarily adapted in practical application by the majority of patients. Still, the fact that participant 6—as one of two participants reporting to follow a gaze pattern during trials and the only participant stating to also follow the gaze pattern in everyday life now—displayed the highest increase in both DVF and navigation performance in the real-world obstacle course suggests further research towards the training of individualized gaze patterns in people with limited VF.

Saccade characteristics such as frequency of saccades, saccade directions, or ratio of exploratory saccades did not show significant effects between training and control phase over all patients. However, unlike the gaze pattern similarity, which was mostly consistent across all

patients, changes in gaze characteristics differ dramatically between patients. Notably, participant 4, who was mentioned already to have displayed unusual DVF results both during training and in the real-world task, was found to continue this trend in the gaze characteristics. Both in saccade frequency and in the ratio of "side-facing" relative saccade angles between 45° and 135° (which can be assumed to be the most effective in scanning the visual area), participant 4 displayed the highest decrease after the training phase and the highest increase after control phase. Similarly, patients with DVF results that suggest positive training effects, such as participant 3 and 6, display mostly positive trends in the gaze characteristics as well.

The influence of patient-related parameters, such as age, previous experience with VR, or experience with other gaze training, can not feasibly be analyzed given the small study population. It can be assumed that younger individuals and those with more VR experience may initially perform better in the gaze training, resulting in a lower entry barrier. However, the effectiveness of training is not expected to be influenced by initial gaze training performance, but rather by the strategies and behavior developed during training. Thus, a study with a much larger population is required to determine whether factors like age and previous VR experience may influence the effectiveness of training positively, negatively, or not at all.

There are several aspects of the gaze training itself that could not be tested within the scope of this project, but hold important research questions for future studies. For one, the design of the training phase did not provide for individual evaluation of the three different tasks. As all tasks were carried out concurrently and real-world task performance was assessed only before and after the full training phase comprised of all three tasks, we cannot draw conclusions about the individual effects that each training task has on real-world navigation performance or gaze behavior. For optimization of the gaze training and to further understand how different virtual tasks can influence real-world performance, the analysis of individual tasks is suggested as a future research topic. Additionally, our study focused on patients with Retinitis pigmentosa to maintain a homogeneous study group and avoid introducing additional variables that could affect the results. Nevertheless, the positive outcomes of our study highly suggest the exploration of the training application in other conditions involving peripheral visual field loss, such as glaucoma or Bardet-Biedl syndrome [44].

The feedback received by participants was overall positive. During the study, six participants (participant 3, 4, 5, 6, 7 and 8) expressed their interest and willingness to continue the gaze training if it becomes available, though participant 8 specified she would prefer to only do the Search Task if training continued. Two participants mentioned to have recommended the training software to friends and acquaintances, and two participants stated to regularly notice improvements in visual and navigation performance in everyday life since the training phase started. Questionnaire results support the overall positive reception of the gaze training, with consistently high ratings in task enjoyment (average 7.26/10) and intuitiveness and ease of use of the software (average 7.85/10). Participants reported some eye strain in the beginning of the training (average 5.86/10) which decreased towards the end of training (average 3.61/10).

To our knowledge, only two other studies were published that investigate the effect of gaze training on the real-world navigation performance in Retinitis pigmentosa patients [14, 15]. It must be noted that quantitative comparison between the results of different gaze trainings is feasible only to a very limited degree due to the different experimental setups in which they were acquired. In their study investigating the effect of a computer display based gaze training application on navigation performance, Ivanov et al. developed a gaze training that consists of an exploratory search task very similar to the Search Task of our work. It was found that a group of RP patients (n = 14) displayed a significant increase in their preferred percentage walking speed by ∼6% in a real-world obstacle course after a six-week training period (total of 15 hours), with no significant improvements in obstacle avoidance in the trials. It was already

suggested by Ivanov et al. that the application of VR devices may impact the effect of gaze training positively. Considering the notably larger improvements found after the VR training —compared to those found after training with a screen-based setup—it can be assumed that VR based gaze training shows higher potential to improve navigation performance in real-world tasks. No numeric performance results are reported by Nguyen et al., who used a very similar training setup to that of Ivanov et al. in a group of n = 14 RP patients. It is reported that significant improvements in navigation performance were found only in patients with VF ≤20˚ diameter. Our results have shown significant improvements in either trial duration or collision avoidance after training in six out of eight patients. The two patients not showing any significant improvements are participants 2 and 4, who have some of the largest measured average VF diameters of the patient group at 18.41˚ and 25.0˚, respectively. This aligns with the findings by Nguyen et al. suggesting that gaze training may be more effective in patients with VF diameter under 20˚. A larger study population is required to statistically validate this hypothesis. Gunn et al. [19] assessed the influence of a short, supervised gaze training consist- ing of two one-hour sessions of general scanning techniques and explicit instructions on opti- mized gaze behavior. The study population consisted of 13 elderly glaucoma patients. The training was found to drastically reduce collisions in a mobility task by up to 88%, with a reduction in walking speed of 10%. Additionally, significant changes in gaze behavior were reported. While the reduction in collisions surpasses the average of 50% reduction found in our study, it has to be considered that the 10% slower movement speed provides patients with more time to plan their walking path and react to obstacles. Furthermore, the study by Gunn et al. lacks a control group, thus it does not distinguish between actual training effects and the improvements in performance that occur naturally from repeating the evaluation task.

One of the most common training methods for low vision patients is Orientation & Mobil- ity (O&M) training. While it is no gaze training, it does fulfill a similar purpose in that it aims to improve walking speed and reduce the number of collisions. Surprisingly, despite the popu- larity of O&M training, controlled studies evaluating its quantitative effects on low vision patients by comparing navigation performance before and after training are scarce. Soong et al. [45] have conducted such a study, testing the navigation performance of 19 elderly patients with varying low vision conditions after one or multiple sessions of supervised, stan- dardized O&M training. However, patients were not found to have significantly improved in either walking speed or collision avoidance directly after the training. Overall, the results found in our study seem promising compared to literature, seeing how—unlike most previous training methods—significant improvements in both walking speed and collision avoidance were found. It is however unclear to which degree this result can be attributed to the training paradigm, rather than to differences in other factors such as evaluation methods, study popula- tion, or training duration.

Despite these promising findings for VR gaze training, the VR based setup also introduces certain limitations: As mentioned in Study population, two participants, in addition to the eight who completed the study, withdrew from the study at an early stage. The primary reason for their discontinuation was directly associated with the use of a VR headset for the training sessions. The first of the two patients reported an increase in migraine attacks when carrying out the training. Virtual Reality is known to cause motion sickness or headaches in some users [46, 47] and it is thus likely that the use of the VR headset did have an effect on the increase in migraine attacks. It was decided to stop the study participation after three training sessions to avoid any risk and discomfort for the participant. The second participant who discontinued the study displayed severe difficulties in navigating within the VR environment when first being introduced to the gaze training. They reported feeling completely disoriented and unable to complete the tasks on their own, and it was thus decided that carrying out the training in an

unsupervised at-home scenario would not be feasible. These two cases highlight that VR gaze training may not be suited for all patients—or, at the very least, requires additional improvements towards the mitigation of motion sickness and the optimization of intuitive tutorials, interfaces, and task design.

A common feedback from participants was that the starting difficulty of the Navigation Task as well as the threshold to advance to higher difficulty levels was too high. Three out of eight participants were not able to advance to a higher difficulty level over the entire duration of the study. They still displayed improvements in performance, but they did not reach the threshold—calculated based on a combination of trial duration and number of collisions—at which the difficulty level would increase. This suggests lowering the starting difficulty of the Navigation Task by decreasing the length of the course as well as the number of obstacles.

Contrary to the Navigation Task, the Target Tracking Task has a low entry difficulty, however participants reported a steep increase in both difficulty and resulting stress at higher difficulty levels. The main factor is the increase of targets that must be tracked simultaneously in order to be successful in the task. Tracking two targets did not prove a big challenge for any of the participants, however once a third target is introduced, difficulty and stress are drastically increased, and none of the participants were able to consistently track more than three targets at the same time. This lead to a stagnation at the difficulty levels around the threshold between three and four marked targets for many participants. To avoid this issue, it may help to limit the number of marked targets to three and instead purely focus on the increase of other difficulty parameters, such as the movement speed of the targets or the area in which targets are free to move around.

Despite these limitations, the developed gaze training shows very promising results, and the use of Virtual Reality as a medium for gaze training seems feasible. Furthermore, it can be emphasized that the training has led to significant improvements in navigation performance despite the VR training itself being fully carried out in seated or stationary standing positions. It can be assumed that the inclusion of real-world mobility would improve training effects further. However, one of our primary goals was to develop a gaze training protocol that could be conveniently and risk-free carried out from home with no need for supervision. We see this as an important measure to enhance user acceptance, especially when considering the practical application of the training beyond controlled research settings. Thus, it was important to show that even with stationary VR training setup, significant improvements in navigation performance in real-world tasks could be achieved. However, the comparison of effects between seated and mobile training conditions provides an interesting research question for subsequent studies, prompting a discussion about the risk-benefit ratio of the inclusion of real-world mobility in Virtual Reality gaze training setups.

The gaze training is currently published as a 'work-in-progress' open-source software on GitHub [48]. This will allow everyone with access to one of the supported VR devices to test and use the gaze training for free once the changes are implemented.

## Conclusion

The results after four weeks of training with the developed gaze training software are promising, showing that Virtual Reality gaze training has the potential to improve the navigation performance of people living with Retinitis pigmentosa in real-world tasks. The majority of participants reported the training software—along with the Virtual Reality device—to be intuitive and easy to use, making it suitable for at-home training with no supervision and with minimal introduction time. However, while VR proves to be a viable medium for a gaze training tool, it can also act as an entry barrier for people being susceptible to motion sickness or people

facing difficulties with orienting and navigating in virtual environments. Still, the developed gaze training shows potential to have a significant positive impact on real-world navigation performance and is currently available as work-in-progress open-source software (link: https://github.com/ANCoral05/VR-GT---Virtual-Reality-Gaze-Training).

## Supporting information

**S1 File. Measurement data.** The raw measurements for the real-world obstacle course, the three gaze training tasks as well as the participant questionnaires.
(XLSX)

**S1 Video. Visual training tasks.** A screen recording of the three visual training tasks of the gaze training.
(ZIP)

**S2 Video. Obstacle course trial.** Example of one of the trials in the real-world obstacle course, captured by the scene camera of the Pupil Labs Invisible eye tracker.
(ZIP)

**S1 Appendix. Additional information regarding the gaze pattern evaluation, VF data and the results of the statistical methods applied in this study.**
(PDF)

**S1 Checklist. CONSORT checklist.**
(PDF)

## Acknowledgments

For her indispensable supervision, guidance, support, and expertise, many thanks go to Enkelejda Kasneci of the Technical University of Munich (TUM). We also thank Gabriele Roever and Stefan Küster of Pro Retina Baden-Württemberg for their advice and aid in acquiring participants for this study. Further gratitude goes to Nadine Wagner of the Tübingen city administration for her assistance in securing a location for the experiment. Lastly, we extend our thanks to all our study participants for their contributions and feedback.

## Author Contributions

**Conceptualization:** Alexander Neugebauer, Iliya Ivanov, Siegfried Wahl.

**Data curation:** Alexander Neugebauer.

**Formal analysis:** Alexander Neugebauer.

**Funding acquisition:** Iliya Ivanov, Siegfried Wahl.

**Investigation:** Alexander Neugebauer, Alexandra Sipatchin.

**Methodology:** Alexander Neugebauer.

**Project administration:** Siegfried Wahl.

**Resources:** Siegfried Wahl.

**Software:** Alexander Neugebauer.

**Supervision:** Siegfried Wahl.

**Visualization:** Alexander Neugebauer.

**Writing – original draft:** Alexander Neugebauer.

**Writing – review & editing:** Alexandra Sipatchin, Katarina Stingl, Iliya Ivanov, Siegfried Wahl.

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
