## [Decision Letter · Decision Letter 0]

5 Oct 2023

PONE-D-23-25461Influence of open-source virtual-reality based gaze training on navigation performance in Retinitis pigmentosa patients in a crossover randomized controlled trialPLOS ONE

Dear Dr. Neugebauer,

Thank you for submitting your manuscript to PLOS ONE. After careful consideration, we feel that it has merit but does not fully meet PLOS ONE’s publication criteria as it currently stands. Therefore, we invite you to submit a revised version of the manuscript that addresses the points raised during the review process.

We look forward to receiving your revised manuscript.

Kind regards,

Antoine Coutrot

Academic Editor

PLOS ONE

Journal Requirements:

4. Thank you for stating the following in the Competing Interests/Financial Disclosure* (delete as necessary) section: 

   "This work was done in an Industry-on-Campus-cooperation between the University of Tübingen and Carl Zeiss Vision International GmbH. Two of the authors, Iliya Ivanov and Siegfried Wahl, are employees of Carl Zeiss Vision International GmbH. Their affiliation with Carl Zeiss Vision International GmbH had no influence in the study. There are no competing interests related to employment, consultancy, patents, products in development, or marketed products. We fully adhere to PLOS ONE policies on sharing data and materials."

We note that one or more of the authors are employed by a commercial company: Carl Zeiss Vision International GmbH 

Additional Editor Comments:

On top of the issues raised by the 2 reviewers, I would add that the figures need to be improved. Bar graphs hide the underlying distribution of the data, see https://garstats.wordpress.com/2016/03/09/one-simple-step-to-improve-statistical-inferences/

Please consider using an alternative such as box plots, violin plots, raincloud plots, or even simply add individual data points.

Reviewers' comments:

Reviewer's Responses to Questions

**Comments to the Author**

1. Is the manuscript technically sound, and do the data support the conclusions?

Reviewer #1: Yes

Reviewer #2: Partly

2. Has the statistical analysis been performed appropriately and rigorously? 

Reviewer #1: Yes

Reviewer #2: Yes

3. Have the authors made all data underlying the findings in their manuscript fully available?

Reviewer #1: Yes

Reviewer #2: Yes

4. Is the manuscript presented in an intelligible fashion and written in standard English?

Reviewer #1: Yes

Reviewer #2: Yes

5. Review Comments to the Author

Reviewer #1: This manuscript describes a VR gaze training protocol for patients experiencing visual field defects due to retinitis pigmentosa. The efforts the authors developed to create and validate the training protocol is commendable. Conclusions from this study has to be put into perspective considering the low number of participants. This is the sad reality when testing such clinical groups, and I understand the pressure to publish their work quickly but I wish the authors could have doubled the participant numbers (still half of what they had initially planned) or considered an age-matched control group.

The authors are missing a review of the literature on gaze training, what has been implemented in the past, the results (in-lab and in the field). This, in order to show what does and doesn't work in gaze training. Also, past studies and encouraging prior findings/results must have influenced the authors' choices when creating their protocol, at least the choice of tasks. Therefore, the authors should frame their work within the body of studies on the subject to justify their implementations and choices.

The authors implemented a complex protocol with different visual tasks. As it stands they cannot say if all tasks contributed to the final performance improvement. I believe it important to have supplementary experimental conditions in which participants would only do part of the tasks to measure the contribution of each. If the authors plan to make recommendations related to gaze training they should be able to pinpoint precisely the contribution power of each of the tasks.

In general, I am surprised to read that the authors only consider RP patients, they should justify what is particular about this population and why it could not have been assimilated into a "peripheral field defect group" with (for example) glaucoma patients to increase the sample number.

Would the authors consider the importance of an age-matched control group? Such a group would allow to establish a performance baseline in all tasks, as well as possibly a higher bound on how much of an increase in performance can be expected from the training protocol. A control group would also permit gaze pattern comparisons, for example to determine if efficient gaze strategies (coping with peripheral field loss) diverge or converge with the control group's behaviour.

Below I list minor comments:

* Method

p.23 - l.91 - how was 30% of a healthy VF proportion calculated? Naïvely, I would do the following: (52*39) / (180*135) = 8.3%.

Figure 2 helps visualise a tunnel vision with a 15° diameter aperture, but considering that the search space is to be 39° vertically it does not seem accurate. Shouldn't the tunnel vision blob be much larger, approximately 40% of the search space height?

l.119 - "The movements direction is controlled via the participant’s body orientation, which is measured through the VR headset’s orientation". How would participants rotate if seating in a chair? Did the authors recommend a swivel chair? Did they gather data about how the participants conducted the training at home? It would be interesting for future reference to know if learning is enhanced by standing or if sitting down is fine.

l.161 - Mention of a time limit in the navigation task is made but no time limit information was provided in the task description (l.116).

l.167 - The use of the term "Scanpath" is awkward. The authors seem to use it to refer to a unique gaze pattern. I advise against using the term "scanpath" in that manner. A scanpath is usually taken to be a sequence of fixation/saccades, the positional information may be what is important, in some cases fixation and saccade features are (duration, amplitudes). But it is not as specific as the authors make it to be. I would recommend "gaze patterns" as an alternative.

l.240 - Could the authors share the details of the power analysis they have run, and why they estimated a 25% performance increase?

l.413 - this part is unclear to me, could the authors please rephrase?

The authors forgot to include information about the nature and frequency of eye tracking calibration for both devices (VR and mobile). This is very important because testing with glasses can significantly decrease the quality of the gaze data and therefore make estimating saccade and fixation features very hard, so special care must have been given to making sure that calibration quality was high.

The authors should give more information about the eye tracking data gathered and how it was processed. For instance, in the case of Pupil Labs device, head rotation data were missing I suppose, therefore gaze parsing was made with missing data (analyses of eye-in-head data only in a head-free task).

In the case of the eye tracker embedded in the VR device, head rotations were available to the authors, they should explain how they processed gaze and if they combined head and eye data together (eye-in-head or eye-in-space).

The authors do not share how VR gaze data was analysed to identify saccades and fixations, this information should be added in the method.

The analyses of fine saccade behaviour may not be possible considering the gaze data quality gathered with this protocol, nevertheless I encourage the authors to consider analyses based on saccade directions. See for example, Fig.3 of https://doi.org/10.1167/jov.22.4.12

Saccade directions can reveal much about visual strategies, for example in terms of return saccades which could decrease as participants learn to more efficiently visually sample their environment.

* Results

Fig.11 - It is unclear why measures related to navigation duration and collisions are superposed.

* Discussion

Related to an earlier comment about a missing literature review in the introduction, studies referenced 13, 25 and 33 are examples of works that appear in the discussion but should also serve to set the context of the study in the introduction.

The authors could consider discussing results of the questionnaire data in the result section instead of touching on them in the discussion only.

The discussion contains details that I consider irrelevant, for example l.722, related to participant drop-out. In my opinion this belong in the method to explain the low participant number, but does not serve any purpose in a discussion.

* Misc.

Ref 36 - dashes in the github address were interpreted as a long dash by LaTeX and invalidated the URL (similar comment for link at the very end of the conclusion).

Reviewer #2: This manuscript describes a virtual-reality-based, at-home gaze training program for people with retinitis pigmentosa (RP). Participants were tested on an obstacle course before and after a control period and the intervention in a crossover design. The results suggest that training is effective in reducing completion time on the obstacle course and obstacle collisions, and increasing the visual area observed by participants. Although the premise of the work is interesting, and there is a need for this type of training for people with RP, a strong rationale for the choice of training and integration with literature is lacking. Furthermore, some of the conclusions are not supported by the results as I understand them. These concerns, along with several others, are summarized below.

Major concerns:

1. The scientific motivation behind the gaze training program is lacking in the introduction. Although there is limited research on gaze training, the authors should provide details on the current published techniques. It would be worthwhile to include details on studies related to healthy older adults (e.g., Young and Hollands 2010), training of medical skills (e.g., Vine SJ, et al. 2013), more details on the cited work in glaucoma (i.e., Gunn et al. 2019), and visual impairments more broadly (e.g., Kuyk T et al. 2010), to name a few. I recognize that the authors are studying RP, but it is important to establish the rationale for their work, and studies in other populations provide that information and motivation. The authors discuss exploratory saccades. While exploratory saccades may be related to screen-based tasks, it is well known that gaze is highly task-relevant during walking/navigation, with very specific relationships between gaze location/timing and foot placement metrics and obstacles. Given the current version of the introduction, it is not clear why the authors chose a generic gaze training protocol (see my subsequent comment as well). The authors should explain, with evidence, why the tasks they chose to affect gaze would help with safe walking.

2. I have two concerns with the gaze training protocol. First, training was done while seated. This is a major limitation given that the goal was to aid in walking/navigation. I recognize that training was at-home and thus, safety is a consideration. However, this major limitation must be acknowledged as such in the discussion section. Gaze training for walking/navigation should ideally be conducted while a person is moving. Second, I am not convinced that the authors “trained” gaze. They did not enforce the scan path (which is somewhat like gridline searches taught by orientation and mobility specialists and is something the authors should consider discussing in the manuscript). In none of the tasks did the authors “teach” or “instruct” participants on how to allocate gaze for better mobility. Thus, I question whether the authors should refer to their intervention as gaze training. The authors must address these concerns and provide a strong rationale if they wish to continue to call their intervention gaze “training”.

3. The authors report changes in trial duration, collisions, and dynamic field of view results after the intervention and after the control period separately. However, the authors state in several places (abstract, lines 551-553, 1st paragraph of discussion, etc.) that these separate results suggest that the training was effective. This conclusion does not seem appropriate if the authors did not directly compare the control and interventions statistically. I could not tell if the authors did a direct comparison from the reported statistics in the appendix. Thus, all the statements mentioned above should be modified or toned down unless direct comparisons were made or the authors can provide a better justification for their statements.

4. The discussion section is lacking. The authors have made very little attempt to integrate their work with previous studies. How does their work compare to other gaze training studies in other eye disease populations and/or healthy populations? Given that glaucoma impairs peripheral vision, it seems like a brief discussion on work in this population would fit at the very least. There is also gaze training work in hemianopia that might fit as well.

Minor concerns:

1. Line 6-7, RP occurs in 1 in 4000 people: that estimate seems very high. Are the authors sure this number is correct?

2. Age varied greatly in this study. Did age play a role in the ability to use the VR as well as performance? One would expect younger participants to handle the VR better and perform better as well. The authors should make mention of any potential age effects (note: I am not looking for the authors to conduct statistical tests with age; rather, some brief comments on whether age might have had an effect is sufficient).

3. In Figure 2, what is the purpose of the tunnel vision images? It is not clear if this was part of the training or related to the manuscript.

4. Lines 177-180: At least some brief rationale should be provided here rather than a reader having to wait until the discussion. This relates to my earlier comment about scan paths.

5. Table 1: please indicate units of measurement, as appropriate (e.g., what units is visual acuity reported in?). In addition, what is VisioCoach?

6. Lines 313-320: Why separate collisions into full and light? Both are collisions and show lack of performance.

7. Dynamic field of view: please provide a citation for this measure. Also, is this the visual area based on visual field and not based on fixation location? Overall, the measure is hard to understand the way it is currently described.

8. Trial duration (or speed) is more a measure of confidence or comfort than showing the effectiveness of gaze training. The fact that trial duration is not a reflection of changes in gaze patterns should be acknowledged in the discussion section.

9. Figure 8: how did the authors get significance for individual participants?

10. Line 567: the change in DFoV of 4.41% doesn’t seem to match what figure 9 shows.

11. Line 583: why is this presented second if this was the main manipulation of the study? The authors should report on the results of the gaze training protocol itself before reporting the results of the obstacle course (that assess the effectiveness of the training).

12. Lines 643-645: the authors cannot make this summary statement given the reported mixed results. Please revise this section.

13. Line 657: the word “here” implies the authors are discussing their own study. However, the paragraph reads as though they are discussing the referenced study. If the latter is correct, then please change “Here, we” to “In that study, they”.

14. Lines 756-771: I am not sure this section is needed. This information is more for the authors than the reader.

6. PLOS authors have the option to publish the peer review history of their article (what does this mean?). If published, this will include your full peer review and any attached files.

Reviewer #1: **Yes: **Erwan DAVID

Reviewer #2: No

---

## [Author Response · Author response to Decision Letter 0]

17 Nov 2023

Response to the editor

Dear Editor,

Thank you for your time and effort and for your support in this submission process! We have revised our manuscript according to your and the reviewers' valuable feedback. The changes made in accordance to your feedback are found below:

- We have changed all titles and subtitles to follow sentence case and have changed an instance in the manuscript from "Tab" to "Table". We have also adjusted the formatting of the corresponding author email and fixed the numbers of the affiliations to now be properly displayed in superscript. Please let us know if there are any additional changes to be made!

- We have adjusted the consent statement to now declare that consent was given in written form. Changes were implemented both in the manuscript and in the submission form.

- We have removed mentions of the funder (DFG) from the acknowledgement section.

4. Thank you for stating the following in the Competing Interests/Financial Disclosure* (delete as necessary) section: 

 "This work was done in an Industry-on-Campus-cooperation between the University of Tübingen and Carl Zeiss Vision International GmbH. Two of the authors, Iliya Ivanov and Siegfried Wahl, are employees of Carl Zeiss Vision International GmbH. Their affiliation with Carl Zeiss Vision International GmbH had no influence in the study. There are no competing interests related to employment, consultancy, patents, products in development, or marketed products. We fully adhere to PLOS ONE policies on sharing data and materials."

We note that one or more of the authors are employed by a commercial company: Carl Zeiss Vision International GmbH 

- We have changed/added the following two statements to the cover letter: "Funding statement: The study was funded by the German Research Foundation (DFG), including support in the form of salary for the author A.N. In addition, the Carl Zeiss Vision International GmbH provided support in the form of salaries for I.I. and S.W. Both funders did not have any additional role in the study design, data collection and analysis, decision to publish, or preparation of the manuscript. The specific roles of these authors are articulated in the ‘author contributions’ section."

"Competing interest statement: This work was done in an Industry-on-Campus-cooperation between the University of Tübingen and Carl Zeiss Vision International GmbH. Two of the authors, I.V. and S.W., are employees of Carl Zeiss Vision International GmbH. This does not alter our adherence to PLOS ONE policies on sharing data and materials. There are no competing interests related to employment, consultancy, patents, products in development, or marketed products."

- We are not aware of any mentions of the ethics statement outside of the methods section. Please let us know if this comment is referring to a specific instance in the manuscript that requires changing.

On top of the issues raised by the 2 reviewers, I would add that the figures need to be improved. Bar graphs hide the underlying distribution of the data, see https://garstats.wordpress.com/2016/03/09/one-simple-step-to-improve-statistical-inferences/

Please consider using an alternative such as box plots, violin plots, raincloud plots, or even simply add individual data points.

- Thank you for this suggestion! We have reworked the plots shown in Figs 8-11 to now show individual samples and overall better represent the underlying data.

Response to reviewers

Dear reviewers, 

We would like to express our gratitude for the thoughtful and constructive feedback that helped us to greatly improve the quality of our work. In this response, we will address each of the points raised and outline changes made to the manuscript.

Thank you for your time and expertise!

General note: In response to one of the comments made, we have decided to rename the Dynamic Field of View (DFoV) variable into Dynamic Visual Field (DVF). For consistency with the reworked manuscript, we will refer to this variable as DVF in this response. Similarly, the term “Scanpath” was substituted with “gaze pattern” and will also be referred to this way within this response.

Regarding the formatting of this response: All reviewer comments are marked in bold text. Replies directly addressed to the reviewers are written in normal font. Text changed or added to the manuscript is written in green font. The lines in which respective changes are found refer to the manuscript version in which changes are marked. [Note: the formatting is only available in the uploaded document, as the form field of the submission form does not support such features.] 

Response to reviewer #1:

Reviewer #1: This manuscript describes a VR gaze training protocol for patients experiencing visual field defects due to retinitis pigmentosa. The efforts the authors developed to create and validate the training protocol is commendable. Conclusions from this study has to be put into perspective considering the low number of participants. This is the sad reality when testing such clinical groups, and I understand the pressure to publish their work quickly but I wish the authors could have doubled the participant numbers (still half of what they had initially planned) or considered an age-matched control group.

The authors are missing a review of the literature on gaze training, what has been implemented in the past, the results (in-lab and in the field). This, in order to show what does and doesn't work in gaze training. Also, past studies and encouraging prior findings/results must have influenced the authors' choices when creating their protocol, at least the choice of tasks. Therefore, the authors should frame their work within the body of studies on the subject to justify their implementations and choices.

- We appreciate your comment! We have expanded the introduction to now include additional literature on gaze training for different types of visual impairments, including those suggested. The section is found from lines 30-75 in the manuscript: “The concept of gaze training for low-vision compensation has been investigated and applied before. Nelles et al. (Nelles, 2001) and Pambakian et al. (Pambakian, 2004) both evaluated the effects of a four-week supervised gaze training in patients with hemianopia, a condition of half-sided visual field loss. In the study of Nelles et al., training included specific instructions for adaptive gaze strategies, whereas patients in the study by Pambakian et al. were free to develop their own gaze strategies. In both studies, it could be shown that after gaze training, patients had a significantly shorter reaction time for visual stimuli in the non-seeing side of the visual field. Additionally, patients reported improvements in several vision-related quality of life aspects after training. Nguyen et al. (Nguyen, 2012), Roth et al. (Roth, 2009) and Ivanov et al. (Ivanov, 2016) conducted studies comprised of six weeks of unsupervised at-home training with a screen-based exploratory saccade training in patients with hemianopia (Roth et al.) and RP (Nguyen et al., Ivanov et al.), respectively. They were assessing the training effect on visual search (Roth et al., Ivanov et al.), scene exploration (Roth et al.), and the effect on real-world mobility (Nguyen et al., Ivanov et al.). Similarly, Kuyk et al. (Kuyk, 2010) investigated the effects of five days of visual search training on both search and real-world mobility tasks in people with different visual field impairments. All three studies with visual search testing paradigm found improvements in reaction time after training, both for digital feature search and for real-world object selection. For the real-world mobility tests, limited effects were reported: Nguyen et al. found significant training effects for real-world navigation in patients with visual field size <10°. In the study by Ivanov et al., RP patients displayed a significant improvement in walking speed, but no improvements in collision avoidance. In the study by Kuyk et al., no significant effects in walking speed were found, but collision avoidance improved in one of the two tested lighting conditions. A different study by Hazelton et al. (Hazelton, 2020) compared the effectiveness of four different eye movement training tools on patients with stroke-induced visual field loss. Quantitatively, no significant improvements were found for any of the four tools, with only individual patients displaying improvements in certain testing paradigms such as visual search or reading speed. Qualitative assessment suggested, however, that patients perceived a positive influence of the training tools on everyday visual tasks. Gunn et al. (Gunn, 2019) conducted a study in which patients with visual impairments caused by glaucoma underwent two supervised one-hour training sessions comprised of both general and task-specific gaze strategy training and instructions, including video showcases of "expert" mobility performers. Effects of the training were evaluated in a foot-placement task and a short obstacle avoidance task, with significant performance improvements found in foot placement accuracy and obstacle avoidance, though at a reduction in movement speed in the obstacle avoidance task. Additionally, changes in the patients' gaze behavior were registered after training. Lastly, Young and Holland (Young, 2010) tested whether gaze training could improve mobility and reduce risk of falling even in elderly persons with no visual field impairment. After a supervised training in which participants received instructions on gaze behavior, participants were found to show increased foot placement accuracy, with no significant changes on movement speed. It can be noted that all of these training paradigms rely on either personal supervision and instructions (Nelles et al., Pambakian et al., Gunn et al., Young and Holland) or use a screen-based setup for at-home training (Nguyen et al., Roth et al., Ivanov et al., Kuyk et al., Hazelton et al.). With the constant advancements in technology and accessibility of Virtual Reality (VR) headsets, a question is raised about the potential of VR to be applied for gaze training purposes.”

In addition, we have implemented minor additions to better convey how the previous literature influenced choices and design decisions made in our work. Lines 113-117: “The training should be usable in an unsupervised at-home environment. The realizability and feasibility of this aim was demonstrated in studies such as by Ivanov et al. (Ivanov, 2016) and Kuyk et al. (Kuyk, 2010).”; Lines 164-166: “Inspired by visual search gaze training methods as applied in different previous studies (Pambakian, 2004; Ivanov, 2016; Roth, 2009), this task requires participants to search an area in front of them for specified visual cues.”

The authors implemented a complex protocol with different visual tasks. As it stands they cannot say if all tasks contributed to the final performance improvement. I believe it important to have supplementary experimental conditions in which participants would only do part of the tasks to measure the contribution of each. If the authors plan to make recommendations related to gaze training they should be able to pinpoint precisely the contribution power of each of the tasks.

- We fully agree that isolating the three visual tasks in order to measure their individual impact on performance will provide valuable insights into potential training optimizations, as well as for the development of specialized training routines. It must be considered, however, that a larger study population is required to individually evaluate all three tasks while maintaining the same statistical power. Thus, we believe that for this study, an individual analysis would not have been feasible. Still, we are interested in assessing these impacts - along with the effects of other training parameters - in future studies, and have added this statement in lines 923-928 of the discussion: ”As all tasks were carried out concurrently and real-world task performance was assessed only before and after the full training phase comprised of all three tasks, we cannot draw conclusions about the individual effects that each training task has on real-world navigation performance or gaze behavior. For optimization of the gaze training and to further understand how different virtual tasks can influence real-world performance, the analysis of individual tasks is suggested as a future research topic.” 

In general, I am surprised to read that the authors only consider RP patients, they should justify what is particular about this population and why it could not have been assimilated into a "peripheral field defect group" with (for example) glaucoma patients to increase the sample number.

- While investigating the impact of VR gaze training on glaucoma patients is an important part of future research, our choice for this project was to maintain homogeneity within the study group. This decision aimed to minimize the variables that would influence the results of the study. Regarding the choice of RP condition over other conditions: Research on Retinitis pigmentosa is a focal point at the Institute for Ophthalmic Research and the University Eye Clinics Tübingen. Previous research that influenced the original design and proposal of this project, such as the study by Ivanov et al. (Ivanov et al., 2016), already focused on the effects of gaze training specifically on RP patients. Considering this, the decision was made to build upon the existing expertise in this research field.

We have added a note in the discussion, lines 929-933, declaring that the investigation of VR gaze training for other types of peripheral visual field loss will be a primary focus of future studies: “Additionally, our study focused on patients with Retinitis pigmentosa to maintain a homogeneous study group and avoid introducing additional variables that could affect the results. Nevertheless, the positive outcomes of our study highly suggest the exploration of the training application in other conditions involving peripheral visual field loss, such as glaucoma or Bardet-Biedl syndrome (Forsythe, 2012).”

Would the authors consider the importance of an age-matched control group? Such a group would allow to establish a performance baseline in all tasks, as well as possibly a higher bound on how much of an increase in performance can be expected from the training protocol. A control group would also permit gaze pattern comparisons, for example to determine if efficient gaze strategies (coping with peripheral field loss) diverge or converge with the control group's behavior.

- Thank you for this suggestion. The inclusion of a reference control group consisting of visually healthy participants was part of the original study protocol. We acknowledge the value of having a baseline for performance, as it would indeed provide a better context for interpreting our results.

However, several factors influenced our decision to ultimately forego the inclusion of this control group. Our experimental tasks, particularly the real-world obstacle course, are not intended to present any significant challenge to visually healthy individuals. Therefore, it is reasonable to assume that a visually healthy group would exhibit very few, if any, collisions during the experiments, and their trial movement speed would closely match their average preferred walking speed.

During the planning stage of the study, we encountered limitations, such as restricted availability of the experimental room as well as general time constraints, which made it apparent that the initial scope of the project would not be realizable within the given timeframe. Acknowledging that the results from a visually healthy control group do not directly address our primary research questions and would likely confirm what can already be assumed with a high degree of certainty, we made the decision to prioritize our time and efforts on the patient group. We will investigate how the experimental setup in future studies can be adjusted to make the test with a visually healthy control group more feasible, also with regards to keeping the tasks interesting and motivating for this group.

Below I list minor comments:

* Method

p.23 - l.91 - how was 30% of a healthy VF proportion calculated? Naïvely, I would do the following: (52*39) / (180*135) = 8.3%.

- Thank you for pointing out the inaccuracy of our phrasing. We have changed the sentence to now specify that the area represents “30% of the visual angles of a healthy VF”. We hope that this clarifies the statement.

Figure 2 helps visualise a tunnel vision with a 15° diameter aperture, but considering that the search space is to be 39° vertically it does not seem accurate. Shouldn't the tunnel vision blob be much larger, approximately 40% of the search space height?

- The dimensions of the search area in both target tracking and search task change with the difficulty level, mentioned in the section ‘Adaptive difficulty levels’. To be precise, the screenshots shown in Figure 2 were taken from a version of the software used for demonstration purposes, where difficulty levels could be manually selected. The search area presented in Figure 2 has dimensions of 80° * 60°. We have added a note to the captions of Figure two to clarify this: “The grey training area has dimensions of 80°×60° in these examples, representing an easy-to-medium difficulty level.”

l.119 - "The movements direction is controlled via the participant’s body orientation, which is measured through the VR headset’s orientation". How would participants rotate if seating in a chair? Did the authors recommend a swivel chair? Did they gather data about how the participants conducted the training at home? It would be interesting for future reference to know if learning is enhanced by standing or if sitting down is fine.

- A swivel chair was recommended, though in our experience any chair without armrest provided sufficient turning space for the task. We have not collected official data regarding seated or standing position of patients. Even if the information was retrospectively acquired, the low number of samples would likely not allow for meaningful results, as it could not be distinguished whether performance differences between patients are caused by their pose or by other factors of inter-patient variability. A study design to test this effect would likely involve asking patients to alternate their pose between every trial, which can be considered for a future study in which it is assessed how varying different aspects of training can influence its effectiveness.

l.161 - Mention of a time limit in the navigation task is made but no time limit information was provided in the task description (l.116).

- Thank you for pointing out this missing information. We have added a more detailed explanation, including the time threshold used for scoring navigation trials, to the description of the navigation task in lines 202-205: “A trial was considered fully successful if the trial duration was below a specified threshold (default 60 seconds). A trial was considered completely failed if the goal was not reached within twice the duration of the threshold. Collisions reduced the threshold by approximately 15% of its current value.”

l.167 - The use of the term "Scanpath" is awkward. The authors seem to use it to refer to a unique gaze pattern. I advise against using the term "scanpath" in that manner. A scanpath is usually taken to be a sequence of fixation/saccades, the positional information may be what is important, in some cases fixation and saccade features are (duration, amplitudes). But it is not as specific as the authors make it to be. I would recommend "gaze patterns" as an alternative.

- Thank you for noting this. We agree that “gaze pattern” is a better choice of term, and have changed all instances in the manuscript accordingly.

l.240 - Could the authors share the details of the power analysis they have run, and why they estimated a 25% performance increase?

- Power analysis was done based on the formula for clinical sample size calculation in longitudinal studies comparing mean change with two time points, as found in Rosner’s Book “Fundamentals of Biostatistics” (https://www.unilus.ac.zm/lms/e-books/books/Basic_Sciences/Behavioural sciences and public health/Fundamentals of Biostatistics (7th Edition).pdf]), Equation 8.30 on page 305. The exact calculation for it is

n_group= (2σ^2*〖(z_(1-α/2)+z_(1-β))〗^2)/Δ^2 

which, given the critical z values z_(1-α/2)=1.96 and z_(1-β)=0.84 that are specified for 0.95 confidence and 0.80 power, standard deviation σ=20%, and effect size Δ=25%, equates to

n_group= (2*20^2*〖(1.96+0.84)〗^2)/25^2 =10.035

We have added additional detail as to which method was used for sample size calculation, as well as where to find it, to the Study population section in lines 315-317: “The sample size was determined following the calculation for sample size in longitudinal studies comparing mean change with two time points, found in Rosner (Rosner, 2011), equation 8.30), […]”.

Regarding the estimation for expected effect size: Given the unique VR-based gaze training setup of this study, an accurate baseline for the expected performance changes in previous literature was lacking. The studies with the most similar paradigms were those by Nguyen et al. (Nguyen, 2012), Ivanov et al. (Ivanov, 2016), Kuyk et al. (Kuyk, 2010), and Gunn et al. (Gunn, 2019), as all of them used a real-world mobility test for evaluation of gaze training. Since both trial duration and number of collisions contribute to the overall navigation performance, we tried to estimate a value that considers both variables. Both Nguyen et al. and Kuyk et al. did not report specific values for improvement in walking speed or collision avoidance. Ivanov et al. and Gunn et al. showed varying effect sizes for mobility-related parameters, from 6% increase in percentage preferred walking speed in the study by Ivanov et al. to a collision reduction of 88% in Gunn et al. Ivanov et al. used a screen-based setup for the training, whereas Gunn et al. used supervised instructed real-world training setups. The Virtual Reality setup applied in our study can be assumed to provide a closer experience to real-world tasks than a screen-based setup, but at the same time would likely not provide the same results as a supervised training fully realized in a real-world environment. Thus, the expected effect size for the general navigation performance was chosen in-between the findings of those studies at 25%. The standard deviation of 20% was based on a study by Baroudi et al. (Baroudi, 2022) investigating walking speed in real-world scenarios, which is noted in lines 317-318: “[…] assuming a standard deviation of 20% (estimated based on Baroudi et al. (Baroudi, 2022)) of the mean navigation performance”.

l.413 - this part is unclear to me, could the authors please rephrase?

- Thank you for making us aware of this. We have improved the sentence structure, now found in lines 557-561: “For the Pre-Post test, a Linear Mixed Model (LMM) with trial duration as dependent variable is used. As a fixed factor, the ‘pre-post condition’ is applied. This binary parameter signifies whether a respective trial originates from the real-world session before or after the relevant phase.”

The authors forgot to include information about the nature and frequency of eye tracking calibration for both devices (VR and mobile). This is very important because testing with glasses can significantly decrease the quality of the gaze data and therefore make estimating saccade and fixation features very hard, so special care must have been given to making sure that calibration quality was high.

- Eye tracking during the real-world obstacle course trials was done using the Pupil Labs Invisible eye tracker, which is connected to a smartphone. No glasses were worn while using this eye tracking device. The eye tracker is calibrated at the start of each session by manually adjusting the offset between gaze direction and a specified target which the wearer is tasked to look at. This calibration method is not very accurate and prone to human error, which is a general limitation of the Pupil Labs Invisible device and has to be considered during experiment design. We decided that this limitation is acceptable for our experimental setup since our gaze evaluation only considers relative gaze movements, no fixations on specific targets or regions. Thus, while we cannot rule out that minor shifts in measured gaze direction may have occurred due to imprecise calibration or changes over the course of the session, these shifts would have little to no influence on any of the measured gaze results.

Regarding VR eye tracking, calibration was done in the in-person session before the start of the training, with patients using the same visual aid they would later use during training. An exception is patient 1, who wore glasses during the in-person session but would later use contact lenses. However, this patient was instructed to repeat the calibration with contact lenses before starting training at home. As stated before, the DVF - which is the only gaze-related parameter measured for the VR gaze training - is not susceptible to influence from shifts in calibration. In addition, the performance and gaze behavior during the VR gaze training have a relatively low priority for the overall research question of this study, as the interest is mostly on results of the real-world trials. Thus, in order to reduce effort for patients as well as the potential for technical complications, patients were not specifically instructed to re-calibrate the device during training.

The authors should give more information about the eye tracking data gathered and how it was processed. For instance, in the case of Pupil Labs device, head rotation data were missing I suppose, therefore gaze parsing was made with missing data (analyses of eye-in-head data only in a head-free task).

- We apologize for the lack of clarity on this topic. Head rotation data is provided by the Pupil Labs Invisible (based on an inertial measurement unit) and was measured individually, as mentioned in the section “Gaze Direction, Dynamic Field of View, and gaze pattern similarity” (now renamed to “Dynamic visual field”), line 583ff. Fig. 9 shows results for both eye-in head (head-centric) and eye-in-space (world-centric) DVF results. We added information on the specific tracking data used for saccade detection in line 440-442: “In a first step, the current gaze speed for each sample frame of the captured eye tracking data was calculated, summing both eye movements and head rotation angles.”

In the case of the eye tracker embedded in the VR device, head rotations were available to the authors, they should explain how they processed gaze and if they combined head and eye data together (eye-in-head or eye-in-space).

It is mentioned in line 501ff. that both head-centered gaze direction and head rotation were measured in all training trials. To calculate the combined gaze direction, the two angles were summed up. For calculating the DVF, the combined gaze direction was used directly on a frame-by-frame basis. The sentence in lines 504-505 was expanded to detail this: ”This data was used to calculate the DVF on a frame-by-frame basis, using the combined gaze direction.” 

In addition, as noted in the previous comment, we specified how the gaze angles were calculated in the real-world condition as well in line 440-442. Thank you for pointing out this lack of information.

The authors do not share how VR gaze data was analysed to identify saccades and fixations, this information should be added in the method.

- Thank you for the feedback. We have added a detailed description of the method to determine saccades, which follows an approach by Nyström et al. Please note that the entire section was expanded in accordance with the following point regarding analysis of fine saccade behavior. As such, the analysis method was reworked and expanded compared to the originally submitted manuscript. The information on saccade detection is found in lines ~438-451: “In addition to the DVF, the eye tracking data were also used to determine saccades during the trial, following an approach by Nyström et al. (Nyström, 2010). In a first step, the current gaze speed for each sample frame of the captured eye tracking data was calculated, summing both eye movements and head rotation angles. The data was smoothed out by applying a moving window average over five frames (~25ms). Next, the mean μ and standard deviation σ of the noise of the eye tracking samples were determined. For this, only samples with angular gaze speed below 100°/s were considered. Following the approach of Nyström et al., a saccade was detected when a peak in the gaze speed surpassed a threshold v_max=μ+6*σ. If a saccade was detected like that, the start- and end point of the saccade were determined based on a second threshold v_onset=μ+3*σ. Appendix D shows an example graphic that visualizes saccade detection using this approach.”

Saccades were not determined for VR gaze data. The DVF was evaluated on a frame-by-frame basis. As mentioned in the previous point, a respective note was added to lines 504-505.

The analyses of fine saccade behaviour may not be possible considering the gaze data quality gathered with this protocol, nevertheless I encourage the authors to consider analyses based on saccade directions. See for example, Fig.3 of https://doi.org/10.1167/jov.22.4.12

Saccade directions can reveal much about visual strategies, for example in terms of return saccades which could decrease as participants learn to more efficiently visually sample their environment.

- Thank you for this suggestion. We have taken the chance to expand our analysis of eye tracking data accordingly. This includes the section on saccade detection reported in the previous point. Furthermore, the following sections were added:

Measurement parameters:

Lines 450-459: “The saccades were analyzed for different characteristics. The first characteristic was the ratio of exploratory saccades, defined as the number of saccades larger than the average visual field radius divided by the total number of saccades per trial. Next, the saccade frequency was determined as the average number of saccades per second. The average ratio between the horizontal and vertical components of saccades was determined to assess whether patients move their gaze more vertically or horizontally. Following an approach by David et al. (David, 2022), we analyzed the direction of each saccade relative to its preceding saccade. This provides additional insight into the patients’ gaze behavior.”

Statistical evaluation: 

Line ~594-596: “The different saccade characteristics - ratio of exploratory saccades, saccade frequency, ratio of vertical to horizontal gaze movements, as well as the change in directions of saccades, were evaluated analogous to the DVF.”

Results: 

Lines 759-768: “Similar to the results of DVF and gaze pattern, a direct comparison between effects of training and control phase on different saccade characteristics does not show significance. Saccade frequency was found to increase by 3.20% after training and decrease by 1.73% after the control phase (p=0.36). Exploratory saccade ratio decreased by 1.32% after training and increased by 0.69% after control (p=0.44). 6.5% more vertical eye movements were displayed after training, 0.39% more after the control phase (p=0.09). No significant changes are found in the direction of saccades relative to the preceding saccade (p=0.068 for angles of 45°-135°, p=0.53 for angles of at least 135°). It can be noted that a strong variation between individual patients was found for the results, which is further elaborated in ‘Discussion’.”

Discussion: 

Lines 899-910: “Saccade characteristics such as frequency of saccades, saccade directions, or ratio of exploratory saccades did not show significant effects between training and control phase over all patients. However, unlike the gaze pattern similarity, which was mostly consistent across all patients, changes in gaze characteristics differ dramatically between patients. Notably, participant 4, who was mentioned already to have displayed unusual DVF results both during training and in the real-world task, was found to continue this trend in the gaze characteristics. Both in saccade frequency and in the ratio of "side-facing" relative saccade angles between 45° and 135° (which can be assumed to be the most effective in scanning the visual area), participant 4 displayed the highest decrease after the training phase and the highest increase after control phase. Similarly, patients with DVF results that suggest positive training effects, such as participant 3 and 6, display mostly positive trends in the gaze characteristics as well.”

* Results

Fig.11 - It is unclear why measures related to navigation duration and collisions are superposed.

- This presentation was chosen to visualize the total performance within the navigation task. However, given that the two parameters are evaluated and reported on separately everywhere else, we agree that this presentation may be confusing. We have adjusted Fig. 11 to now show both parameters as separate bars.

* Discussion

Related to an earlier comment about a missing literature review in the introduction, studies referenced 13, 25 and 33 are examples of works that appear in the discussion but should also serve to set the context of the study in the introduction.

- As stated in our response to the mentioned comment, the studies by Ivanov et al. and Nguyen et al (previously referenced 13 and 33) are described in more detail in the introduction, and statements of the influence of the study by Ivanov et al. on the study design of this work have been made. Since our preliminary work (Neugebauer, 2021) addresses a more technical question regarding gaze training, we have included an additional text passage in the methods section in lines 239-242: “The shape of the gaze pattern was selected following the findings of a previous study (Neugebauer, 2021) in which two popular gaze patterns were tested. The gaze pattern that was suggested to patients in this study (Fig 4) was found to lead to better results in both navigation as well as search tasks when compared to the competing pattern.”

The authors could consider discussing results of the questionnaire data in the result section instead of touching on them in the discussion only.

- Prompted by your feedback, we have included the average standard deviation for the individual question-and-task conditions of the questionnaire in Fig. 12 and have addressed these results in the following section in lines 795-801: “The questionnaire ratings show high standard deviation between participants, with scores oftentimes ranging from 1 to 10 within the same conditions. Some of the scores of individual patients do not align with verbal feedback given after the study and may thus be a result of misinterpretation of the question. Still, these consistently high standard deviations indicate that the different aspects of the training tasks, such as motivation, perceived difficulty, and enjoyment, are highly subjective.”

The discussion contains details that I consider irrelevant, for example l.722, related to participant drop-out. In my opinion this belong in the method to explain the low participant number, but does not serve any purpose in a discussion.

- Thank you for your feedback. We agree that the patient drop-out should be mentioned in the methods section. However, we believe that the topic also holds value for the discussion, as it highlights VR-related issues that may pose barriers for other patients as well and should thus be considered in the overall evaluation of VR as a gaze training tool. To better highlight that this is the conclusion drawn from the section, we have added a sentence in lines 1009-1012: “These two cases highlight that VR gaze training may not be suited for all patients - or, at the very least, requires additional improvements towards the mitigation of motion sickness and the optimization of intuitive tutorials, interfaces, and task design.” We have also expanded the Study population section in line 309f to reference these additional details: “10 patients […] participated in the study, two of which discontinued the study early on. Details about the reasons for discontinuation are provided in the Discussion.”

* Misc.

Ref 36 - dashes in the github address were interpreted as a long dash by LaTeX and invalidated the URL (similar comment for link at the very end of the conclusion).

- Thank you very much, we have fixed the links to now be compiled as URLs properly.

Response to reviewer #2

Reviewer #2: This manuscript describes a virtual-reality-based, at-home gaze training program for people with retinitis pigmentosa (RP). Participants were tested on an obstacle course before and after a control period and the intervention in a crossover design. The results suggest that training is effective in reducing completion time on the obstacle course and obstacle collisions, and increasing the visual area observed by participants. Although the premise of the work is interesting, and there is a need for this type of training for people with RP, a strong rationale for the choice of training and integration with literature is lacking. Furthermore, some of the conclusions are not supported by the results as I understand them. These concerns, along with several others, are summarized below.

Major concerns:

1. The scientific motivation behind the gaze training program is lacking in the introduction. Although there is limited research on gaze training, the authors should provide details on the current published techniques. It would be worthwhile to include details on studies related to healthy older adults (e.g., Young and Hollands 2010), training of medical skills (e.g., Vine SJ, et al. 2013), more details on the cited work in glaucoma (i.e., Gunn et al. 2019), and visual impairments more broadly (e.g., Kuyk T et al. 2010), to name a few. I recognize that the authors are studying RP, but it is important to establish the rationale for their work, and studies in other populations provide that information and motivation. The authors discuss exploratory saccades. While exploratory saccades may be related to screen-based tasks, it is well known that gaze is highly task-relevant during walking/navigation, with very specific relationships between gaze location/timing and foot placement metrics and obstacles. Given the current version of the introduction, it is not clear why the authors chose a generic gaze training protocol (see my subsequent comment as well). The authors should explain, with evidence, why the tasks they chose to affect gaze would help with safe walking.

- We appreciate your comment! We have expanded the introduction to now include additional literature on gaze training for different types of visual impairments, including those suggested: Lines 30-75: “The concept of gaze training for low-vision compensation has been investigated and applied before. Nelles et al. (Nelles, 2001) and Pambakian et al. (Pambakian, 2004) both evaluated the effects of a four-week supervised gaze training in patients with hemianopia, a condition of half-sided visual field loss. In the study of Nelles et al., training included specific instructions for adaptive gaze strategies, whereas patients in the study by Pambakian et al. were free to develop their own gaze strategies. In both studies, it could be shown that after gaze training, patients had a significantly shorter reaction time for visual stimuli in the non-seeing side of the visual field. Additionally, patients reported improvements in several vision-related quality of life aspects after training. Nguyen et al. (Nguyen, 2012), Roth et al. (Roth, 2009) and Ivanov et al. (Ivanov, 2016) conducted studies comprised of six weeks of unsupervised at-home training with a screen-based exploratory saccade training in patients with hemianopia (Roth et al.) and RP (Nguyen et al., Ivanov et al.), respectively. They were assessing the training effect on visual search (Roth et al., Ivanov et al.), scene exploration (Roth et al.), and the effect on real-world mobility (Nguyen et al., Ivanov et al.). Similarly, Kuyk et al. (Kuyk, 2010) investigated the effects of five days of visual search training on both search and real-world mobility tasks in people with different visual field impairments. All three studies with visual search testing paradigm found improvements in reaction time after training, both for digital feature search and for real-world object selection. For the real-world mobility tests, limited effects were reported: Nguyen et al. found significant training effects for real-world navigation in patients with visual field size <10°. In the study by Ivanov et al., RP patients displayed a significant improvement in walking speed, but no improvements in collision avoidance. In the study by Kuyk et al., no significant effects in walking speed were found, but collision avoidance improved in one of the two tested lighting conditions. A different study by Hazelton et al. (Hazelton, 2020) compared the effectiveness of four different eye movement training tools on patients with stroke-induced visual field loss. Quantitatively, no significant improvements were found for any of the four tools, with only individual patients displaying improvements in certain testing paradigms such as visual search or reading speed. Qualitative assessment suggested, however, that patients perceived a positive influence of the training tools on everyday visual tasks. Gunn et al. (Gunn, 2019) conducted a study in which patients with visual impairments caused by glaucoma underwent two supervised one-hour training sessions comprised of both general and task-specific gaze strategy training and instructions, including video showcases of "expert" mobility performers. Effects of the training were evaluated in a foot-placement task and a short obstacle avoidance task, with significant performance improvements found in foot placement accuracy and obstacle avoidance, though at a reduction in movement speed in the obstacle avoidance task. Additionally, changes in the patients' gaze behavior were registered after training. Lastly, Young and Holland (Young, 2010) tested whether gaze training could improve mobility and reduce risk of falling even in elderly persons with no visual field impairment. After a supervised training in which participants received instructions on gaze behavior, participants were found to show increased foot placement accuracy, with no significant changes on movement speed. It can be noted that all of these training paradigms rely on either personal supervision and instructions (Nelles et al., Pambakian et al., Gunn et al., Young and Holland) or use a screen-based setup for at-home training (Nguyen et al., Roth et al., Ivanov et al., Kuyk et al., Hazelton et al.). With the constant advancements in technology and accessibility of Virtual Reality (VR) headsets, a question is raised about the potential of VR to be applied for gaze training purposes.”; Lines ~95-98: “However, research on the use of Virtual Reality for adjusting gaze behavior in other fields, such as for industry task training (Harris, 2021), medical procedures (Vine, 2013), or as therapeutic intervention for patients with mental health disorders (Selaskowski, 2023), suggests that the use of VR applications is feasible to influence gaze behavior.”

Regarding the design choices for the three training tasks, we have added multiple sections: Lines ~138-148: “Considering the unsupervised nature of the training, it wasn't practical to base the training on specific gaze instructions given to patients, as is typically done in supervised experimental training conditions (Nelles, 2001; Gunn, 2019). While patients could have received instructions before training, continuously monitoring patients over the course of the training to ensure that instructions are followed correctly would not have been possible. Acknowledging this, the training tasks were instead designed such that their success criteria naturally promote exploratory saccades and frequent eye movements, encouraging patients to develop own strategies and adaptive behavior. This follows the approach of previously mentioned studies by Nguyen et al. (Nguyen, 2012), Ivanov et al. (Ivanov, 2016), or Pambakian et al. (Pambakian, 2004).”; Lines 164-166: “Inspired by visual search gaze training methods as applied in different preliminary studies (Pambakian, 2004; Ivanov, 2016; Roth, 2009), this task requires participants to search an area in front of them for specified visual cues.”

2. I have two concerns with the gaze training protocol. First, training was done while seated. This is a major limitation given that the goal was to aid in walking/navigation. I recognize that training was at-home and thus, safety is a consideration. However, this major limitation must be acknowledged as such in the discussion section. Gaze training for walking/navigation should ideally be conducted while a person is moving. Second, I am not convinced that the authors “trained” gaze. They did not enforce the scan path (which is somewhat like gridline searches taught by orientation and mobility specialists and is something the authors should consider discussing in the manuscript). In none of the tasks did the authors “teach” or “instruct” participants on how to allocate gaze for better mobility. Thus, I question whether the authors should refer to their intervention as gaze training. The authors must address these concerns and provide a strong rationale if they wish to continue to call their intervention gaze “training”.

- Thank you for your feedback! We have added a comment in lines 122-124 to initially acknowledge this seemingly contradictory decision: “The implications of this sacrifice of real-world mobility - in a training specifically designed to improve the mobility of patients - will be further addressed in the discussion.”

We would like to emphasize that the choice of a seated training environment was a deliberate one: One of our primary goals was to develop a gaze training protocol that could be conveniently and risk-free carried out from home with no need for supervision. Importantly, despite the seated training environment, our study demonstrated a significant increase in navigation performance in a real-world setting after completing the training. This outcome strongly suggests that the training had the intended effect on walking and navigation skills. We have added a section in lines 1035-1047 of the revised manuscript to address this topic: “Furthermore, it can be emphasized that the training has led to significant improvements in navigation performance despite the VR training itself being fully carried out in seated or stationary standing positions. It can be assumed that the inclusion of real-world mobility would improve training effects further. However, one of our primary goals was to develop a gaze training protocol that could be conveniently and risk-free carried out from home with no need for supervision. We see this as an important measure to enhance user acceptance, especially when considering the practical application of the training beyond controlled research settings. Thus, it was important to show that even with stationary VR training setup, significant improvements in navigation performance in real-world tasks could be achieved. However, the comparison of effects between seated and mobile training conditions provides an interesting research question for subsequent studies, prompting a discussion about the risk-benefit ratio of the inclusion of real-world mobility in Virtual Reality gaze training setups.”

Regarding the second part of your comment: We understand the concerns regarding the use of the term “gaze training”. It is correct that the training does not involve any direct instructions for mandatory gaze pattern tasks or specific gaze behavior. This, in parts, is again a result of the aim to develop this VR software as a tool adapted for independent and unsupervised training, as it can be difficult to evaluate and enforce the correct execution of such explicit instructions without direct supervision. Instead, the visual tasks applied in this study were designed for a more “gamified” training approach. The success criteria for these tasks naturally encourage participants to make increased eye movements. This “instruction-less” gaze training paradigm is in line with previous studies, such as the ones by Pambakian et al. (Pambakian, 2004), Nguyen et al. (Nguyen, 2012), and Ivanov et al. (Ivanov, 2016). The validity and effectiveness of this approach is demonstrated by the results of the Dynamic Visual Field analysis in Fig. 11 of the manuscript. In this analysis, all patients exhibited a noticeable improvement in DVF across nearly all tasks, indicating that the training tasks effectively foster changes in eye movements. While it can’t be clearly stated that these effects on gaze behavior found within the virtual environment transfer to real-world tasks (as is discussed in the next comment), it is evident that the tasks can have a notable influence on the patients’ gaze behavior. Considering this, we believe that the term "gaze training" remains an accurate description for the developed tool, despite its lack of any direct teaching methods or explicit instructions. We hope that the section in lines 138-148, mentioned in the previous comment, sufficiently addresses this topic and explains the reasoning behind the chosen instruction-free gaze training method. 

3. The authors report changes in trial duration, collisions, and dynamic field of view results after the intervention and after the control period separately. However, the authors state in several places (abstract, lines 551-553, 1st paragraph of discussion, etc.) that these separate results suggest that the training was effective. This conclusion does not seem appropriate if the authors did not directly compare the control and interventions statistically. I could not tell if the authors did a direct comparison from the reported statistics in the appendix. Thus, all the statements mentioned above should be modified or toned down unless direct comparisons were made or the authors can provide a better justification for their statements.

- Thank you for raising our awareness to this issue. We have revised and expanded our statistical analysis to incorporate a direct comparison of the result parameters of the real-world tasks. The improvements in trial duration and collision avoidance after training were found to be significantly higher than the improvements after the control phase. However, while individual patients still show notable increase in DVF, the average increase of the training condition was not found to be significantly higher compared to the control condition. Thus, we have revised several of the statements made in this regard, including the Abstract, Statistical methods, Results, Discussion, and Conclusion. Additionally, indicators for significance between the two conditions were added in Fig. 8.

Abstract: 

“On average, the time required to move through the obstacle course decreased by 17.0% after the training phase, the number of collisions decreased by 50.0%. Both effects are significantly higher than those found in the control phase (p<0.001 for required time, p=0.0165 for number of collisions), with the required time decreasing by 5.9% and number of collisions decreasing by 10.4% after the control phase. The average visual area observed by participants increases by 4.41% after training, however the effect is not found to be significantly higher than in the control phase (p = 0.394).” 

Statistical methods:

Lines 538-552: “In the real-world obstacle course tasks, effects for the four measurement parameters were tested. Each parameter was tested using two different paradigms: First, the effects of training- and control phase were assessed individually by testing the data acquired in the session before the respective phase against data acquired in the session after the phase (Pre-Post test). Second, the two effect sizes from training and control phase are tested directly against each other to determine if training effects significantly surpass effects of the control phase (Training-Control test). For this test, delta values for each trial are calculated: For example, the difference between the first trial of the session before the training/control phase and the first trial of the session after the training/control phase is calculated. This way, the effects for each phase can be expressed as a set of delta values. By testing the set of delta values from the training phase against the delta values from the control phase, statistical significance between effect sizes can be evaluated. The following sections describe the statistical models and pre-processing steps for each measurement parameter.”

Lines 567-570: “The Training-Control test was tested mostly analogous to the Pre-Post test, again using an LMM. Delta trial duration was used as dependent variable, with the respective phase (training or control) as fixed factor.”

Lines 580-582 : “For the Training-Control test, values were no longer zero-inflated, as the delta values could be both positive and negative. Thus, a Generalized Linear Mixed Model (GLMM) was applied.”

Lines 585-589: “The DVF was found to follow normal distribution quite well […], allowing the use of an LMM for both Pre-Post test and Training-Control test with no additional transformations. Analogous to the other measurement parameters, the Pre-Post test uses absolute DVF values as dependent variable and the pre-post condition as fixed factor, whereas the Training-Control test uses delta values of the DVF as dependent variable and the respective phase as fixed effect.”

Results: 

Lines 710-723: “After the training phase, participants displayed a significant improvement in trial duration by 17.0% compared to the performance before the training (p<0.001), decreasing the average trial duration from 37.2 (±12.3) seconds to 30.9 (±8.68) seconds. The average number of collisions per trial decreased by 50.0% after training (p<0.001), from 0.513 collisions per trial to 0.256 collisions per trial. A comparison with the results before and after the control phase shows that the training phase was significantly more effective in improving the average trial duration (p<0.001) and reducing the number of collisions (p=0.0165) than the control phase. The average trial duration had improved by 5.9% after the control phase, from 34.8 (±12.7) seconds to 32.7 (±9.87) seconds. The average number of collisions per trial improved by 10.4% after the control phase, from 0.391 to 0.350 collisions per trial. Overall, the results suggest that the training phase was significantly more effective in improving navigation performance compared to the control phase.”

Lines 737-741: “Although the average increase in world-centric DVF of 4.41% is found to be significant (p<0.001) when evaluating the data before and after training, the effect is not significantly larger than the increase in DVF displayed after control (p=0.394) at 2.06%. Three of eight participants (1,3,6) display a notable increase in world-centric DVF after the training phase, with two participants (4,8) showing decreases.”

Discussion: 

Lines 820-822: “While notable changes in DVF are found in individual patients, the group's average DVF increase after training was not found to significantly surpass the effect of the control phase.”

Conclusion:

Line 1066: changed the statement “[…] showing that Virtual Reality gaze training has the potential to improve the visual and navigation performance […]” to now only include navigation performance, as the effect of the training on DVF and other gaze characteristics remains unclear.

4. The discussion section is lacking. The authors have made very little attempt to integrate their work with previous studies. How does their work compare to other gaze training studies in other eye disease populations and/or healthy populations? Given that glaucoma impairs peripheral vision, it seems like a brief discussion on work in this population would fit at the very least. There is also gaze training work in hemianopia that might fit as well.

- We appreciate this feedback. We have added and expanded the discussion of three additional studies (Nguyen, 2012; Gunn, 2019; Soong, 2001) to put the results of our study into perspective. The respective section is found in lines 960-995 in the Discussion: “No numeric performance results are reported by Nguyen et al., who used a very similar training setup to that of Ivanov et al. in a group of n=14 RP patients. It is reported that significant improvements in navigation performance were found only in patients with VF ≤20° diameter. Our results have shown significant improvements in either trial duration or collision avoidance after training in six out of eight patients. The two patients not showing any significant improvements are participants 2 and 4, who have some of the largest measured average VF diameters of the patient group at 18.41° and 25.0°, respectively. This aligns with the findings by Nguyen et al. suggesting that gaze training may be more effective in patients with VF diameter under 20°. A larger study population is required to statistically validate this hypothesis. Gunn et al. (Gunn, 2019) assessed the influence of a short, supervised gaze training consisting of two one-hour sessions with general scanning techniques and explicit instructions on optimized gaze behavior. The study population consisted of 13 elderly glaucoma patients. The training was found to drastically reduce collisions in a mobility task by up to 88%, with a reduction in walking speed of 10%. Additionally, significant changes in gaze behavior were reported. While the reduction in collisions surpasses the average of 50% reduction found in our study, it has to be considered that the 10% slower movement speed provides patients with more time to plan their walking path and react to obstacles. Furthermore, the study by Gunn et al. lacks a control group, thus it does not distinguish between actual training effects and the improvements in performance that occur naturally from repeating the evaluation task.

One of the most common training methods for low vision patients is Orientation & Mobility (O&M) training. While it is no gaze training, it does fulfill a similar purpose in that it aims to improve walking speed and reduce the number of collisions. Surprisingly, despite the popularity of O&M training, controlled studies evaluating its quantitative effects on low vision patients by comparing navigation performance before and after training are scarce. Soong et al. (Soong, 2001) have conducted such a study, testing the navigation performance of 19 elderly patients with varying low vision conditions after one or multiple sessions of supervised, standardized O&M training. However, patients were not found to have significantly improved in either walking speed or collision avoidance directly after the training. Overall, the results found in our study seem promising compared to literature, seeing how - unlike most previous training methods - significant improvements in both walking speed and collision avoidance were found. It is however unclear to which degree this result can be attributed to the training paradigm, rather than to differences in other factors such as evaluation methods, study population, or training duration.”

We also note that results of our preliminary study regarding gaze patterns are further supported by findings of Nelles et al., though the experimental setups are different. Lines 869-874: “The study concluded that the gaze pattern has the potential to enhance visual performance in the presence of tunnel vision and suggests that introducing the gaze pattern in gaze training for individuals with tunnel vision could have beneficial effects. This is in line with a study by Nelles et al. (Nelles, 2001), in which hemianopia patients were introduced to a similar gaze pattern in supervised training, displaying significant improvements in visual search after training.”

Minor concerns:

1. Line 6-7, RP occurs in 1 in 4000 people: that estimate seems very high. Are the authors sure this number is correct?

- All literature we have found on the worldwide prevalence of RP suggests numbers in the range of 1:3000 to 1:5000, with the majority reporting it as 1:4000. We are not aware of any sources that report a widely differing prevalence. We have referenced an additional source (Cross, 2022) to further support this number.

2. Age varied greatly in this study. Did age play a role in the ability to use the VR as well as performance? One would expect younger participants to handle the VR better and perform better as well. The authors should make mention of any potential age effects (note: I am not looking for the authors to conduct statistical tests with age; rather, some brief comments on whether age might have had an effect is sufficient).

- Thank you for this feedback. We have added a respective statement in the discussion, lines 911-919: “The influence of patient-related parameters, such as age, previous experience with VR, or experience with other gaze training, cannot feasibly be analyzed given the small study population. It can be assumed that both a younger age as well as more experience with VR correlate with better initial performance within the gaze training, resulting in a lower entry barrier. However, the effectiveness of training is not expected to be influenced by initial gaze training performance, but rather by the strategies and behavior developed during training. Thus, a study with a much larger population is required to determine whether factors like age and previous VR experience may influence the effectiveness of training positively, negatively, or not at all.”

3. In Figure 2, what is the purpose of the tunnel vision images? It is not clear if this was part of the training or related to the manuscript.

- The purpose of the displayed tunnel vision showcase is purely for demonstration purposes for the manuscript. We have re-phrased the captions of figure 2 to state more clearly that the tunnel vision simulation was not present during the study and thus had not influence on the results: “The tunnel vision simulation is added for visualization purposes in this manuscript only and was not present during participants' training.”

4. Lines 177-180: At least some brief rationale should be provided here rather than a reader having to wait until the discussion. This relates to my earlier comment about scan paths.

- The feedback is appreciated. We have revised the section ‘Suggested gaze pattern’ in lines 239-250 to explain the rationale behind this decision: “The shape of the gaze pattern was selected following the findings of a previous study (Neugebauer, 2021) in which two popular gaze patterns were tested. One of the gaze patterns that were suggested to patients in this study (Fig. 4) was found to lead to better results in both navigation as well as search tasks when compared to the competing pattern. However, findings of this and other gaze training studies (Gunn, 2019) also suggest that training and application of specific, mandatory gaze patterns can potentially result in a reduction of the subjects' walking speed. Thus, in this study, patients were given autonomy in choosing if - and to which degree - they want to follow the suggested gaze pattern. The pattern was visually introduced to the patients on a screen prior to the use of the gaze training, explaining its background and potential advantages.”

5. Table 1: please indicate units of measurement, as appropriate (e.g., what units is visual acuity reported in?). In addition, what is VisioCoach?

- Visual acuity in table 1 is provided as a decimal score. Similar to the two other notations - fraction and LogMAR - the visual acuity is given in reference to “normal” sight and thus does not include any units. A visual acuity of 0.1 is equivalent to a vision of 20/200 in fraction notation or +1.0 in LogMAR. We have added a footnote to table 1 specifying that visual acuity is given as decimal value: “Visual acuity is notated as decimal score.”

Thank you for making us aware of the missing information regarding VisioCoach. We have added an additional footnote to table 1: “A commercially available screen-based gaze training software for RP patients [Odilia, 2023], applied in a study by Ivanov et al. [Ivanov, 2016] and Roth et al. [Roth, 2009] and evaluated by Hazelton et al. [Hazelton, 2020].”

6. Lines 313-320: Why separate collisions into full and light? Both are collisions and show lack of performance.

- The rationale for distinguishing between light and full collisions is based on two factors: 1) Experience from previous studies with similar setups have indicated that participants may occasionally collide with obstacles despite clear intentions to avoid them, indicating that the collisions can be attributed to false estimation of an obstacle’s height or position, rather than a complete lack of awareness of the obstacle. 2) In a real-world scenario, only full collisions would pose a major risk of harm or accidents, whereas light collisions would rarely have negative consequences. As such, participants may be less focused on avoiding light collisions, possibly even accepting them for the sake of faster movement speed or more comfortable movement (e.g. not lowering or turning the head enough to fully avoid a hanging obstacle).

For these reasons, we think that a separation between full and light collisions can potentially provide meaningful information, for example if patients would have displayed no change in the number of collisions overall, but a change in the ratio of full and light collisions. We have added a comment stating this in line 402-408: “This distinction was made to acknowledge that patients may limit their efforts on the avoidance of collisions that would - in a real-life scenario - pose risks of accidents or injuries. Given that light collisions would usually not pose such risks, it is possible that patients put less priority on avoiding these types of collisions. Thus, if a shift from 'mostly full collisions' to 'mostly light collisions' is detected between two sessions, it can be considered a positive effect on navigation performance.” While our study results did not show any notable effects in this regard - the ratio between full and light collisions remaining largely constant - we believe that maintaining this separation offers added granularity and insight into the participants' performance. 

7. Dynamic field of view: please provide a citation for this measure. Also, is this the visual area based on visual field and not based on fixation location? Overall, the measure is hard to understand the way it is currently described.

- Dynamic Field of View/DVF is a custom parameter that was introduced and defined by us as a means to more accurately report on the visual area observed by participants. Parameters such as saccade frequency and amplitude, exploratory saccade ratio, or average gaze speed per second all fail to accurately represent the actual visual area observed by participants. For example, a tunnel vision patient that constantly alternates their gaze between exactly two fixation points would likely display a large saccade frequency, large ratio of exploratory saccades, and a high average gaze speed. However, the gaze movement is rather inefficient, because the majority of the visual surroundings would remain unobserved. The DVF is designed to recognize this and is instead calculated based on the actual area observed over a specified time interval.

The visual area measured to calculate the DVF is based on all eye tracking samples, meaning that samples both during saccades as well as during fixations are considered. Literature suggests that the human eye is capable of detecting visual stimuli even during saccades (https://www.sciencedirect.com/topics/biochemistry-genetics-and-molecular-biology/saccadic-suppression), and as such, it can be assumed that a target or obstacle could be perceived at all times during visual exploration. We have specified this rationale in lines 435-437 of the ‘Measurement parameter’ section: “DVF is calculated considering all eye tracking samples, both during fixations and during saccades, since detection of potential obstacles or points of interest is possible even during eye movements (Robinson, 2022).”

Thank you for the feedback regarding the lack of clarity in our explanation of the DVF parameter. We have made multiple adjustments to improve it: First, as stated in the introduction of our response, we have decided to rename the parameter to “dynamic visual field”. We believe that this name better reflects its definition and also brings it in line with the term “visual field” (VF) that is frequently used in the manuscript (instead of “Field of View”, which can be used synonymously, but may be associated with cameras and displays rather than eyes). Additionally, we have re-phrased the section in lines 409-437 to more clearly explain what the dynamic visual field represents and how it is calculated: “Using Pupil Labs Invisible eye tracking glasses, both the direction of gaze relative to the head and the orientation of the head itself are measured during the real-world obstacle course trials. Based on these two parameters, the dynamic visual field (DVF) is calculated. We define the DVF as the visual area observed over a specified amount of time. When fixating a single point, a person with tunnel vision would only be able to observe the area within their normal VF - in this context, this could be described as the 'static' visual field. However, as soon as the person starts moving their gaze, they will automatically explore and observe new areas of their visual surroundings. Measuring this observed area over a fixed duration results in the DVF. Notably, the DVF only increases if new visual area is explored that has not already been observed within the specified time frame. In this work, the time frame for the DVF is set to three seconds. This means that the DVF at any point of time is defined as the observed visual area over the last three seconds. Averaging the DVF for all measured samples within a trial provides the average DVF for that trial. The DVF is reported and evaluated as a percentage change, showing how much the DVF increased or decreased over the course of the training or control phase. An increase of average DVF of 10% indicates that the person was able to observe 10% more of their surroundings.”

8. Trial duration (or speed) is more a measure of confidence or comfort than showing the effectiveness of gaze training. The fact that trial duration is not a reflection of changes in gaze patterns should be acknowledged in the discussion section.

- We have expanded the section in lines 840-842 of the discussion to now state this more clearly: “A lack of increase in DVF - or even a decrease in DVF - in a patient does not imply the absence of improvements in performance, showing that improved trial duration and collision avoidance are not a direct indicator for changed gaze behavior.”

9. Figure 8: how did the authors get significance for individual participants?

- Significance for individual participants was assessed by individually running the statistical models on data sets consisting only of trials from the respective patient. For these analyses, random factor has no influence. We have added this information to the captions of figure 8: “P-values for individual patients were evaluated by applying the statistical models to a subset of the data containing only trials of the respective patient.”

10. Line 567: the change in DFoV of 4.41% doesn’t seem to match what figure 9 shows.

- The DVF increase of 4.41% refers to the world-space DVF, which is shown in the top graph of figure 9. We have confirmed that the bar indicating the average DVF increase was correctly sized. The distance between two grid lines (indicating a 10% step) was 34 pixels. The height of the DVF bar was 15 pixels, which results in a ratio of exactly 0.441. However, following the editor’s suggestions for our manuscript, result figures were reworked into box plots, thus the previous bar plot in figure 9 no longer exists.

11. Line 583: why is this presented second if this was the main manipulation of the study? The authors should report on the results of the gaze training protocol itself before reporting the results of the obstacle course (that assess the effectiveness of the training). 

- While the VR gaze training was the main manipulation of the study, the results obtained within it are only relevant for secondary evaluations. For the main research question - whether VR based gaze training can significantly improve navigation performance of RP patients in a real-world setting - the results of the real-world trials hold much more weight. While different arguments can be made regarding the structure in which the results are presented, we think that it makes sense to lead with the results most relevant to the main hypothesis, following it up with results related to secondary research questions.

12. Lines 643-645: the authors cannot make this summary statement given the reported mixed results. Please revise this section.

- Thank you for your feedback! We understand that the phrasing of the sentence may be unclear, and in addition may falsely imply that the statement would be statistically validated. Thus, we have rephrased the sentence in lines 839-845 to present the statement more clearly and accurately: “In summary, it is possible that an increased DVF has a positive effect on navigation performance, indicated by the fact that all three patients displaying such increase in DVF also display improvements in navigation performance. However, the opposite statement cannot be made: A lack of increase in DVF - or even a decrease in DVF - in a patient does not imply the absence of improvements in performance, showing that improved trial duration and collision avoidance are not a direct indicator for improved gaze behavior. This further suggests the presence of other factors through which gaze training influences navigation performance.”

13. Line 657: the word “here” implies the authors are discussing their own study. However, the paragraph reads as though they are discussing the referenced study. If the latter is correct, then please change “Here, we” to “In that study, they”.

- Thank you, we have rephrased the sentence as suggested.

14. Lines 756-771: I am not sure this section is needed. This information is more for the authors than the reader.

- Thank you for the feedback, we agree that this section holds little relevance for readers and have thus removed it from the manuscript.

---

## [Decision Letter · Decision Letter 1]

9 Jan 2024

Influence of open-source virtual-reality based gaze training on navigation performance in Retinitis pigmentosa patients in a crossover randomized controlled trial

PONE-D-23-25461R1

Dear Dr. Neugebauer,

We’re pleased to inform you that your manuscript has been judged scientifically suitable for publication and will be formally accepted for publication once it meets all outstanding technical requirements. Please also have a look at R1's last minor comments.

Kind regards,

Antoine Coutrot

Academic Editor

PLOS ONE

Additional Editor Comments (optional):

Reviewers' comments:

Reviewer's Responses to Questions

**Comments to the Author**

1. If the authors have adequately addressed your comments raised in a previous round of review and you feel that this manuscript is now acceptable for publication, you may indicate that here to bypass the “Comments to the Author” section, enter your conflict of interest statement in the “Confidential to Editor” section, and submit your "Accept" recommendation.

Reviewer #1: All comments have been addressed

Reviewer #2: All comments have been addressed

2. Is the manuscript technically sound, and do the data support the conclusions?

Reviewer #1: Yes

Reviewer #2: Yes

3. Has the statistical analysis been performed appropriately and rigorously? 

Reviewer #1: Yes

Reviewer #2: Yes

4. Have the authors made all data underlying the findings in their manuscript fully available?

Reviewer #1: Yes

Reviewer #2: Yes

5. Is the manuscript presented in an intelligible fashion and written in standard English?

Reviewer #1: Yes

Reviewer #2: Yes

6. Review Comments to the Author

Reviewer #1: I wish to thank the authors for addressing all of my comments.

I still believe that it is unfortunate that the authors did not test more RP participants from the start, and did not include an age-matched control group. I understand the practical limitations for that, and I believe that studies such as the one described in this manuscript are too uncommon to reject on the basis of low participant numbers.

I would just ask for the authors to give a more practical and detailed description (with equations if necessary) about the process used for calculating the DVF (in the appendix possibly), so that it becomes easier to replicate.

Reviewer #2: (No Response)

7. PLOS authors have the option to publish the peer review history of their article (what does this mean?). If published, this will include your full peer review and any attached files.

Reviewer #1: **Yes: **Erwan David

Reviewer #2: No

---

## [Editor Report · Acceptance letter]

18 Jan 2024

PONE-D-23-25461R1 

PLOS ONE

Dear Dr. Neugebauer, 

I'm pleased to inform you that your manuscript has been deemed suitable for publication in PLOS ONE. Congratulations! Your manuscript is now being handed over to our production team.

Kind regards, 

on behalf of

Dr. Antoine Coutrot 

Academic Editor

PLOS ONE